# What's In My Big Data?

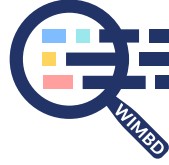

**Yanai Elazar**[1,2]   **Akshita Bhagia**[1]   **Ian Magnusson**[1]   **Abhilasha Ravichander**[1]
**Dustin Schwenk**[1]   **Alane Suhr**[3]   **Pete Walsh**[1]   **Dirk Groeneveld**[1]   **Luca Soldaini**[1]
**Sameer Singh**[4]   **Hannaneh Hajishirzi**[1,2]   **Noah A. Smith**[1,2]   **Jesse Dodge**[1]

[1]Allen Institute for AI
[2]Paul G. Allen School of Computer Science & Engineering, University of Washington
[3]University of California, Berkeley   [4]University of California, Irvine

✉ yanaiela@gmail.com   ⌂ https://github.com/allenai/wimbd   ⊕ wimbd.apps.allenai.org

## Abstract

Large text corpora are the backbone of language models. However, we have a limited understanding of the content of these corpora, including general statistics, quality, social factors, and inclusion of evaluation data (contamination). In this work, we propose What's In My Big Data? (WIMBD), a platform and a set of sixteen analyses that allow us to reveal and compare the contents of large text corpora. WIMBD builds on two basic capabilities—count and search—*at scale*, which allows us to analyze more than 35 terabytes on a standard compute node. We apply WIMBD to ten different corpora used to train popular language models, including *C4*, *The Pile*, and *RedPajama*. Our analysis uncovers several surprising and previously undocumented findings about these corpora, including the high prevalence of duplicate, synthetic, and low-quality content, personally identifiable information, toxic language, and benchmark contamination. For instance, we find that about 50% of the documents in *RedPajama* and *LAION-2B-en* are duplicates. In addition, several datasets used for benchmarking models trained on such corpora are contaminated with respect to important benchmarks, including the Winograd Schema Challenge and parts of GLUE and SuperGLUE. We open-source WIMBD's code and artifacts to provide a standard set of evaluations for new text-based corpora and to encourage more analyses and transparency around them.

## 1 Introduction

Data is the foundation upon which machine learning (ML) is built. The introduction of new datasets drives progress, playing a crucial role in facilitating research and the creation of models with novel capabilities. Over time, the computational cost of AI experiments has dramatically increased, partly due to training increasingly large models on increasingly large datasets (Schwartz et al., 2020; Sevilla et al., 2022); today, some of the most impactful datasets are being created by scraping text from the entire publicly-available internet (Raffel et al., 2020; Together Computer, 2023; Penedo et al., 2023; Soldaini et al., 2024). These are some of the largest text datasets that have ever been built, and they are typically introduced with only a description of how they were made but no documentation of their contents. This is an important distinction, as we are now training models on massive text corpora without knowing what ideas, topics, toxicity, or personal information they contain.

Meanwhile, language models (LMs) have become ubiquitous and are used by people worldwide daily. These AI systems directly impact people's lives, and thus, it has become vitally important to understand their capabilities and drawbacks. Models are only capable of learning from the data they were trained on, but analysis of pretraining corpora is hindered by lack of public release and by their massive size. Work analyzing the contents of web-scale corpora typically focuses on a subset of important dimensions, and there has been almost no work analyzing multiple datasets across the same dimensions. This means that ML practitioners have no practical tools to describe differences between datasets before choosing which one(s) to use.

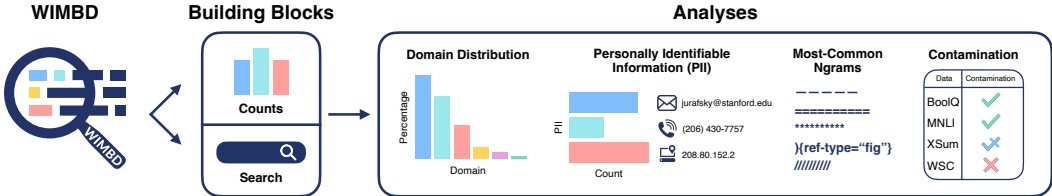

Figure 1: An overview of WIMBD. We implement two fundamental capabilities: *Count* and *Search*, allowing quick processing and access to large text corpora, which enables a wide range of analyses.

In this work, we propose to investigate the content of large text corpora using WHAT'S IN MY BIG DATA (WIMBD), a set of tools that enables practitioners to easily explore and quickly analyze large language datasets. We also use this tool to provide some of the first measurements across different web-scale datasets that are directly comparable. WIMBD has two components: (1) a **search** tool that enables programmatic access to search for documents containing a query using an *Elasticsearch*[1] (ES) index. ES is a search engine that allows retrieving strings from a corpus, the documents where they appeared, and the number of times they appeared. (2) a **count** functionality, built using map-reduce (Dean & Ghemawat, 2008), allowing quick iteration over an entire dataset and extraction of relevant information, e.g., the character length distribution of documents, duplicates, domain counts, finding personally identifiable information (PII), and more. WIMBD is extendable and can be used to index, count, and analyze other corpora at scale (we benchmark the runtimes in Appendix D).

Using these tools, we perform a set of sixteen analyses on ten different English corpora used to train LMs, including *C4* (used to train T5; Raffel et al., 2020), *The Pile* (used to train Pythia; Gao et al., 2020; Biderman et al., 2022; 2023), and *RedPajama* (used to reproduce Llama, Touvron et al., 2023, and to train RedPajama-INCITE; Together Computer, 2023). We divide our analyses into four categories: (1) data statistics (e.g., number of tokens and domain distribution; §4.2); (2) data quality (e.g., most frequent $n$-grams and measuring duplicate documents; §4.3); (3) community- and society-relevant measurements (e.g., benchmark contamination and personally identifiable information detection; §4.4); and (4) cross-corpora analysis (e.g., comparing the most common $n$-gram and document overlap; §B.4). An illustration of WIMBD is presented in Figure 1.

Our work presents many insights on data distribution and anomalies. For example, inspecting the distribution over document lengths exposes anomalies where specific lengths are overrepresented relative to neighboring lengths; these anomalies often correspond to near-duplicate template-generated text or documents arbitrarily truncated to a specific character length. As another example, punctuation sequences are frequently the most common $n$-grams, such as a dash ('−') repeated ten times as the most common 10-gram in *The Pile*. WIMBD offers both retrospective documentation and grounding of model behavior to their training data and actionable insights for higher-quality corpora curation.

## 2 BACKGROUND: ON THE IMPORTANCE OF DATA UNDERSTANDING

There have been repeated calls for ML practitioners to provide better data documentation (e.g., McMillan-Major et al., 2023; Bender & Friedman, 2018; Mitchell et al., 2023; Pistilli et al., 2023; Paullada et al., 2021; Gebru et al., 2021). On the other hand, some of the most impactful ML models are increasingly opaque, specifically with respect to the most important component of recent advancements: data. With the increasingly competitive nature of the field, developers of systems like GPT-4 (OpenAI, 2023) and PaLM-2 (Google, 2023) have been offering little transparency into the most important development decisions, including the sources, size, and contents of their training data.

As web-scale datasets drive this rapid progress in modern ML systems, the gap between data transparency and documentation is more striking than ever (Kaddour et al., 2023). From a technical standpoint, the massive size of these datasets makes analysis of their contents challenging; even if OpenAI or Google shared their training data, it's unclear where to start understanding it in its entirety. Tools like the Data Measurements Tool (Luccioni et al., 2021) and Know Your Data (Google, 2021) work towards improving data documentation, but focus on smaller datasets since the scale of web data leads to significant technical challenges. Our work aims to address this critical missing component.

---

[1]https://www.elastic.co/elasticsearch/

While other works support indexing and analyses of large corpora (Piktus et al., 2023a; Marone & Van Durme, 2023; Simig et al., 2022; Piktus et al., 2023b; Razeghi et al., 2022b), these efforts support a single corpus and often do not support programmatic access to the data or the analysis. Instead, we offer a holistic approach that combines search and counting with a package that allows programmatic access through wrappers on top of the ES API and extendable efficient counting capabilities.

Additional efforts are concerned with the effect of data on model behavior. Longpre et al. (2023) investigate how the composition of LMs' pretraining data influences their downstream performance. Razeghi et al. (2022a) measure high correlation between term frequency and LMs' few-shot reasoning capabilities with those terms. Shin et al. (2022) study the effect of pretraining corpora on in-context abilities. Seshadri et al. (2023) demonstrate that text-to-image models mimic biases from their training data. Akyurek et al. (2022) study fact tracing for identifying pretraining examples that enable a factual assertion, while Guu et al. (2023) offer a training run simulator, which allows making counterfactual queries on what a model would have learned under a different training procedure. These efforts separately built dedicated infrastructure to perform the studies. Our work provides a dedicated interface and tooling that allows performing a wide range of analyses on large-scale corpora, categorizing and offering novel analyses that highlight new insights into these large corpora.

## 3 WIMBD: THE PLATFORM

A core desideratum of WIMBD is to enable quick processing of terabytes of data. As such, we focus on uncomplicated, standard methods from the information retrieval and data management communities. WIMBD is comprised of two basic components: *counting* and *search* (retrieval). Fast counting and retrieving enable us to answer fundamental questions about data, as we demonstrate in Section 4. We summarize the

Table 1: Summary of the capabilities WIMBD provides and the analyses enabled by them.

| Basic Ability | Analyses |
|---|---|
| *Exact Counts* (§3.1) | Document Counts, min/max doc length, #tokens, domain distribution, utterance date statistics, geolocation, language distribution, length distribution, toxic language, personally identifiable information, demographic sentiment co-occurrences |
| *Compressed Counts* (§3.1) | Duplicates, most & least common $n$-grams |
| *Search* (§3.2) | Benchmark contamination, $n$-gram counts |

framework abilities and types of analyses in Table 1. We run our experiments using a compute node machine with 224 CPUs and 882GB RAM, and an Elasticsearch cluster for the indexed corpora.

### 3.1 COUNTING

Due to the sparsity of language data and the scale of the data of interest, accurate counting can be challenging. We leverage the map-reduce framework (Dean & Ghemawat, 2008). We provide two approaches for counting, described below.

**Exact Counts** The exact counts approach is designed for cases where the number of possible values is tractable and can fit in memory. This fits cases where we are interested in calculating a bound number of variables of interest (e.g., number of documents,§4.2, or document length, §4.3.3).

**Compressed Counts** The compressed counts approach is designed for cases where the number of possible values is intractable. For instance, the total 10-grams in a large corpus can be very high, and the memory usage to compute all of them would be overwhelming. Similarly, finding duplicates requires keeping and comparing the strings of all documents in memory. In the case of *C4*, that would require over 800 GB of RAM. Instead, we apply a compression function (e.g., hashing, Bloom, 1970) to those values, reducing memory footprint while sacrificing some accuracy (due to hash collisions). For example, when finding the most common 10-grams, we store a table of counts where the keys in the table correspond to hashes of 10-grams. The hash table size is configurable according to the amount of memory available. The larger the hash table, the smaller the probability of hash collisions and, therefore, the higher the accuracy of the counts. E.g., unigram estimates are more accurate than 10-gram estimates since the number of possible values is much smaller.

### 3.2 SEARCHING

The second part of WIMBD allows fast text retrieval. For instance, we can get the number of documents mentioning a word or sequence (document frequency). It also allows more complex Boolean queries. While search and retrieval have numerous implementations, such as reverse indices, suffix arrays,

Table 2: Summary statistics of the corpora, along with the models trained on them. * signifies that the model was not trained on the exact version we consider, either due to some data mismatch, or the original data being private.

| Corpus | Origin | Model | Size (GB) | # Documents | # Tokens | max(# Tokens) | min(# Tokens) |
|--------|--------|-------|-----------|-------------|----------|---------------|---------------|
| OpenWebText | Gokaslan & Cohen (2019) | GPT-2* (Radford et al., 2019) | 41.2 | 8,005,939 | 7,767,705,349 | 95,139 | 128 |
| C4 | Raffel et al. (2020) | T5 (Raffel et al., 2020) | 838.7 | 364,868,892 | 153,607,833,664 | 101,898 | 5 |
| mC4-en | Chung et al. (2023) | umT5 (Chung et al., 2023) | 14,694.0 | 3,928,733,374 | 2,703,077,876,916 | 181,949 | 1 |
| OSCAR | Abadji et al. (2022) | BLOOM* (Scao et al., 2022) | 3,327.3 | 431,584,362 | 475,992,028,559 | 1,048,409 | 1 |
| The Pile | Gao et al. (2020) | GPT-J/Neo & Pythia (Biderman et al., 2023) | 1,369.0 | 210,607,728 | 285,794,281,816 | 28,121,329 | 0 |
| RedPajama | Together Computer (2023) | LLaMA* (Touvron et al., 2023) | 5,602.0 | 930,453,833 | 1,023,865,191,958 | 28,121,329 | 0 |
| S2ORC | Lo et al. (2020) | SciBERT* (Beltagy et al., 2019) | 692.7 | 11,241,499 | 59,863,121,791 | 376,681 | 1 |
| peS2o | Soldaini & Lo (2023) | - | 504.3 | 8,242,162 | 44,024,690,229 | 97,043 | 154 |
| LAION-2B-en | Schuhmann et al. (2022) | Stable Diffusion* (Rombach et al., 2022) | 570.2 | 2,319,907,827 | 29,643,340,153 | 131,077 | 0 |
| The Stack | Kocetkov et al. (2023) | StarCoder* (Li et al., 2023) | 7,830.8 | 544,750,672 | 1,525,618,728,620 | 26,298,134 | 0 |

suffix trees for exact match search, and dense retrieval for fuzzy search, in this work, we use ES, an inverted index. We build a wrapper on top of the ES API, allowing tailored and customized searches to fit our analysis requirements. We leave it to future work to explore other search alternatives.

## 4    WIMBD: THE ANALYSES

This section presents analyses conducted in WIMBD, grouped by category. First, we describe the ten corpora considered in this study (§4.1). We then consider four high-level categories, each split into several analyses: data statistics (§4.2), data quality (§4.3), and community- and society-relevant measurements (§4.4). Cross-corpus analyses, as well as elaborations and more analyses are presented in the appendix (§B). Our analyses are inspired by previous works (Dodge et al., 2021; Gao et al., 2020), but we expand them to multiple corpora, extend the types of analyses, and open-source our modular toolkit to encourage researchers to scrutinize their corpora. We offer the first extensive analyses on ten, combining extension of previous analyses and several novel ones.

### 4.1    CORPORA

We cover ten different large corpora, spanning across text-only (e.g., *C4*) to image captions (*LAION-2B-en*) and code (*The Stack*). These corpora have been used in training language models (or similar large-scale models, such as Stable Diffusion; Rombach et al. 2022). A high-level description of these datasets using WIMBD is presented in Table 2, and further details about the construction and origin of these corpora are detailed in Appendix A.

### 4.2    DATA STATISTICS

> **Main Findings**
>
> • Four out of the ten corpora we consider have 'empty' documents (meaning they contain only space-like characters), while *The Pile* and *RedPajama* contain the same longest document (with over 28 million tokens) of an encyclopedia.
>
> • While the most common source of webpages in *C4* originates from www.nytimes.com, it consists of less than 0.05% of the total web pages, *mC4-en* most common domain is google.com (over 5% of the documents), and cdn.shopify.com contributes almost 6% to the total documents in *LAION-2B-en*.

#### 4.2.1    SUMMARY STATISTICS

We begin by computing some summary statistics and present the results in Table 2. Using the `Exact Counts` we compute the following high-level statistics of a corpus: (1) size, (2) number of documents, (3) number of tokens,[2] (4) the size of the longest document, and (5) the size of the shortest document. Out of all corpora, *mC4-en* is the largest, which takes 14.7TB of disk, and 2.7 trillion tokens. After that comes *The Stack* with a size of 7.8TB, and more than 1.5 trillion tokens. Interestingly, four corpora contain documents with empty strings: *LAION-2B-en* (81 total), which typically contain a sequence of white spaces. In *The Stack* (1,350 total), *RedPajama* (3,877), and *The*

---

[2]We use Unicode text segmentation (Unicode, 2023) as a tokenizer, but we support any tokenizer supported by HuggingFace's *tokenizers* library (Moi & Patry, 2023).

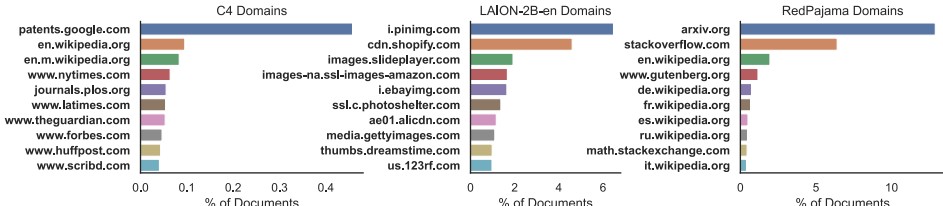

Figure 2: Domain distribution of the ten most common domains per token for *C4*, *LAION-2B-en*, and *RedPajama*.

*Pile* (7,533), documents typically contain a mix of special characters that denote spacing (e.g., '\n', or '\t'). In *RedPajama*, all of the empty strings are from the arXiv subset. The longest document in *The Stack* is a json file, with 26,298,134 tokens from http://jquery.com/. The longest document in *The Pile* and *RedPajama* is the same encyclopedia book called "INTERNATIONAL ENCYCLOPEDIA OF THE SOCIAL & BEHAVIORAL SCIENCES" from the Books3 subset with 28,121,329 tokens.

### 4.2.2 INTERNET DOMAIN DISTRIBUTION

Some corpora contain metadata information about the URL where the documents came from. As such, we employy the Exact Counts functionality, to parse the entire corpus, and extract information from the URLs about the (1) schemas (e.g., *http*, *https*), (2) domains (e.g., www.google.com, en.wikipedia.org, etc.), and (3) suffixes (e.g., *com*, *org*, *de*, etc.).

We apply these counts on the corpora that contain this information, namely *C4*, *mC4-en*, *OSCAR*, *RedPajama*, and *LAION-2B-en*. Starting with the domain analysis, we perform these counts twice: once when each domain is counted per document (yielding documents per domain) and another where each domain is counted per token (yielding tokens per domain). We present the results of three corpora per token in Figure 2 (and the full results in Appendix B.1). First, we note that *C4* contains documents from a diverse set of domains, and even the percentage of the most common one, patents.google.com, is less than 0.05%. On the other hand, in the case of *LAION-2B-en*, cdn.shopify.com is responsible for more than 6% of the documents. Similarly, arxiv.org is responsible for more than 12% of the documents in *RedPajama*. We showcase the results of the domains for the other corpora, as well as the schemas and suffixes in Appendix B.1.

### 4.3 DATA QUALITY

> **Main Findings**
>
> • The most common $n$-grams often correspond to repeated punctuation marks and duplicates.
> • While more than 60% of documents in *The Pile* are duplicates (unsurprisingly due to oversampling), *RedPajama* and *LAION-2B-en* also contain about 50% duplicate documents.
> • Document length distribution reveals interesting (and unexpected) outliers of documents, often resulting from duplicate documents and idiosyncratic data decisions.

### 4.3.1 MOST & LEAST COMMON $n$-GRAMS

Measuring outliers can reveal interesting insights about a corpus (Mitchell et al., 2023), We explore the most and least common token $n$-grams of each corpus using the Compressed Counts . We compute the 10K most common $n$-grams for all corpora, with $n \in \{1, 2, 3, 10\}$. We report the results of the ten most common 10-grams in Table 3 and of the ten most common uni-, bi-, and tri-grams in Table 9 in the Appendix. Identical $n$-grams across corpora are highlighted in the same colors.

The different corpora contain a lot of uncleaned html or markdown format (e.g., ten times '?' or 'amp'), or boilerplate texts such as: ".  You can follow any responses to this entry through" in *C4*, or "( Log Out / Change ) You are commenting using" in *OSCAR*, and formatting ("[1][2][3][") in *S2ORC* and *peS2o*, which signifies references.

A striking finding from this analysis is the vast repetition of such 10-grams. For instance, '?', '.', and '–' repeated ten times appear 9, 7.2, and 4.4 million times, respectively, in *C4*. We perform a manual analysis on the repeating question marks in *C4* to better understand the scenarios where they

Table 3: Most common 10-grams in five of the corpora we consider. $n$-grams from the top-10 that occur in more than one corpus are highlighted in the same color.

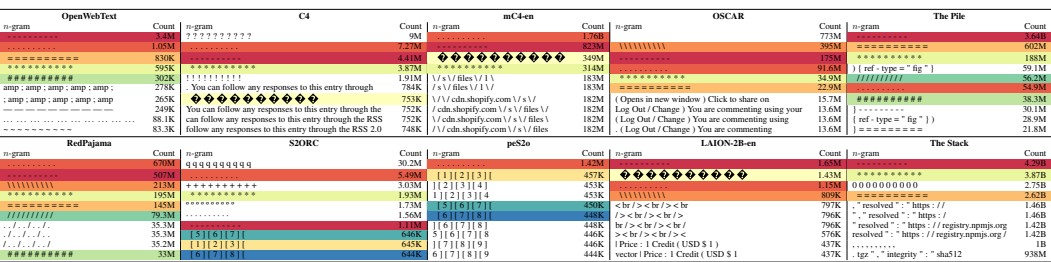

| OpenWebText | | C4 | | mC4-en | | OSCAR | | The Pile | |
|---|---|---|---|---|---|---|---|---|---|
| n-gram | Count | n-gram | Count | n-gram | Count | n-gram | Count | n-gram | Count |
| - - - - - - - - - - | 3.4M | ? ? ? ? ? ? ? ? ? | 9M | - - - - - - - - - - | 1.76M | - - - - - - - - - - | 773M | = = = = = = = = = = | 3.64B |
| . . . . . . . . . . | 1.05M | . . . . . . . . . . | 7.27M | . . . . . . . . . . | 823M | . . . . . . . . . . | 395M | = = = = = = = = = | 602M |
| = = = = = = = = = = | 830K | - - - - - - - - - - | 4.41M | ◆◆◆◆◆◆◆◆◆◆ | 349M | \\\\\\\\\\ | 175M | * * * * * * * * * * | 188M |
| * * * * * * * * * * | 595K | * * * * * * * * * * | 3.87M | * * * * * * * * * * | 314M | . . . . . . . . . . | 91.6M | ) [ ref - type = " fig " ] | 59.1M |
| # # # # # # # # # | 302K | ! ! ! ! ! ! ! ! ! | 1.91M | * * * * * * * * * | 183M | * * * * * * * * * * | 54.9M | / / / / / / / / / / | 56.2M |
| amp ; amp ; amp ; amp ; | 278K | . You can follow any responses to this entry through | 784K | / s \ / files \ / 1 \ / | 183M | = = = = = = = = = = | 22.9M | = = = = = = = = = | 54.9M |
| ; amp ; amp ; amp ; amp ; | 265K | ◆◆◆◆◆◆◆◆◆◆ | 753K | \ / \ / cdn.shopify.com \ / s \ / | 182M | ( Opens in new window ) Click to share on | 15.7M | # # # # # # # # # # | 38.3M |
| — — — — — — — — — | 249K | You can follow any responses to this entry through | 752K | / cdn.shopify.com \ / s \ / files \ / | 182M | Log Out / Change ) You are commenting using your | 13.6M | ] - - - - - - - - - | 30.1M |
| . . . . . . . . . . . . | 88.1K | can follow any responses to this entry through the RSS | 752K | \ / cdn.shopify.com \ / s \ / files \ | 182M | ( Log Out / Change ) You are commenting using | 13.6M | { ref - type = " fig " ] ) | 28.9M |
| - - - - - - - - - - - | 83.3K | follow any responses to this entry through the RSS 2.0 | 748K | / \ / cdn.shopify.com \ / s \ / files | 182M | . ( Log Out / Change ) You are commenting | 13.6M | ] = = = = = = = = = | 21.8M |

| RedPajama | | S2ORC | | peS2o | | LAION-2B-en | | The Stack | |
|---|---|---|---|---|---|---|---|---|---|
| n-gram | Count | n-gram | Count | n-gram | Count | n-gram | Count | n-gram | Count |
| - - - - - - - - - - | 670M | ¶ ¶ ¶ ¶ ¶ ¶ ¶ ¶ ¶ | 30.2M | - - - - - - - - - - | 1.42M | - - - - - - - - - - | 1.65M | - - - - - - - - - - | 4.29B |
| . . . . . . . . . . | 507M | . . . . . . . . . . | 5.49M | ] [ 1 ] [ 2 ] [ 3 ] [ | 457K | * * * * * * * * * * | 1.43M | * * * * * * * * * * | 3.87B |
| \ \ \ \ \ \ \ \ \ \ | 213M | + + + + + + + + + + | 3.03M | ] [ 2 ] [ 3 ] [ 4 ] | 453K | ◆◆◆◆◆◆◆◆◆◆ | 1.13M | 0 0 0 0 0 0 0 0 0 0 | 2.75B |
| * * * * * * * * * * | 195M | * * * * * * * * * * | 1.93M | 1 ] [ 2 ] [ 3 ] [ 4 | 453K | \ \ \ \ \ \ \ \ \ \ | 809K | = = = = = = = = = = | 2.62B |
| = = = = = = = = = = | 145M | ° ° ° ° ° ° ° ° ° ° | 1.73M | [ 5 ] [ 6 ] [ 7 ] [ | 450K | < br / > < br / > < br | 797K | . " resolved " : " https : // | 1.46B |
| . / . . / . . / . . | 79.3M | | 1.56M | [ 6 ] [ 7 ] [ 8 ] [ | 448K | / > < br / > < br / > | 796K | " , " resolved " : " https : // | 1.46B |
| . / . . / . . / . . | 35.3M | [ 5 ] [ 6 ] [ 7 ] [ | 646K | 5 ] [ 6 ] [ 7 ] [ 8 | 446K | br / > < br / > < br / > | 796K | " resolved " : " https : // registry.npmjs.org | 1.42B |
| . / . . / . . / . / | 35.2M | [ 1 ] [ 2 ] [ 3 ] [ | 645K | ] [ 7 ] [ 8 ] [ 9 ] | 446K | > < br / > < br / > < | 576K | resolved " : " https : // registry.npmjs.org / | 1.42B |
| # # # # # # # # # # | 33M | [ 6 ] [ 7 ] [ 8 ] [ | 644K | 6 ] [ 7 ] [ 8 ] [ 9 | 444K | | Price : 1 Credit ( USD $ 1 ) | 437K | ] = = = = = = = = = | 1B |
| | | | | | | vector | Price : 1 Credit ( USD $ 1 | 437K | . - tgz " , " integrity " : " sha512 | 938M |

appear on the ten consecutive question marks symbols and categorize each appearance into *writing*, *noise*, and *format* occurrence. Analyzing 100 random documents, we found that 68% of documents use such $n$-grams as part of their *writing* style (e.g., `... $6???????????  How is that possible?`, or `...  So what do u think????????????????????????`). 18% are due to *noise* as we could not understand the context or content of the writing (e.g., `...  e ???????????????  kap chit-koa ??`), and finally, 14% of the documents were due to different *format* styles or issues (e.g., a sequence of question marks following by a 'normal' text, or a sequence of question marks between keywords).

### 4.3.2 DUPLICATES

Previous work has found that duplication can affect the quality of pretraining data, impacting sample efficiency (Lee et al., 2022; Tirumala et al., 2023) and memorization (Carlini et al., 2023). While more recent work finds contradictory evidence on data with less web-scraped text (Biderman et al., 2023), measuring duplication in pretraining data is necessary for future research on its effects. We calculate duplicates by matching documents with an MD5 hash of their texts (using *Compressed Counts* ). If more than a single document has the same hash, we consider them duplicates.[3] We examine the duplication of document text and URLs within each dataset. While some datasets explicitly deduplicate their content, others do not, and some even oversample some sources.

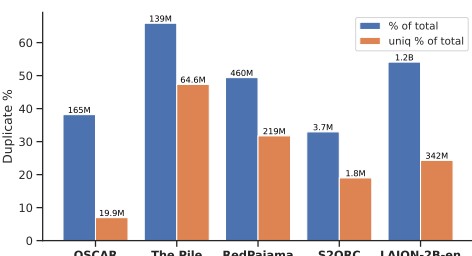

Figure 3: Percentages of document and document cluster duplicates in corpora with $> 1\%$ documents duplicated (corresponding to blue and orange bars). Duplicate counts are above bars.

Table 4: Most frequent text duplicates from four datasets with text duplicates, along with their counts. Truncation for visualization is marked by [...].

| Corpus | Text |
|---|---|
| OSCAR Count: **1.8M** | In order to login you must be registered. Register ing takes only a few moments but gives you increas[...] |
| The Pile Count: **3.8K** | {\n "info" : {\n "version" : 1,\n "author" : "xcode"\n }\n} |
| RedPajama Count: **213.9K** | ACCEPTED\n\n#### According to\nInternational Pla nt NamesIndex\n\n#### Published in\nnull\n\n#### Original n[...] |
| LAION-2B-en Count: **1M** | Front Cover |

In Figure 3 we show counts and ratios of duplication across datasets with greater than 1% documents duplicated, and all datasets are shown in Table 13 in the appendix. These are based on two kinds of counts: (1) the count of documents in all clusters of duplicate text (in blue) and (2) the count of duplicate clusters (in orange). As expected, deduplicated corpora such as *C4* have no exact duplicates (as those were filtered out of the corpus). In contrast, *The Pile*, which intentionally oversampled some data sources, has many duplicates (139M documents belonging to 64.6M duplicate text clusters). *LAION-2B-en* has the second highest ratio of duplicate documents (1.25B documents belonging to 342M duplicate text clusters), perhaps due to the smaller space of short sentences common in

---

[3]To test for hash collisions, we rerun the analysis with a different random seed. None of the $> 7$ billion hashes across the ten corpora had a different count. This could only occur if an identical number of collisions conflated an identical set of counts or, more likely, there were no collisions.

its image "alt text" source. Figure 15 in the appendix showcase the images of the most common duplicates in *LAION-2B-en*, with the most common images describe mainly receipts.

Table 4 showcases duplicates with the most occurrences in four corpora. These duplicates vary dramatically in length and domain. *LAION-2B-en*, *OSCAR*, and *RedPajama* have clusters with the most occurrences, in the hundreds of thousands and above. Top duplicates in *LAION-2B-en* are shorter and describe products and website features. *OSCAR*'s top duplicates are all instances of website boilerplate.[4] *RedPajama*'s top duplicates come from similar templated citation information.

### 4.3.3 Document length distribution

We compute document length distributions with Exact Counts. We expect a smooth distribution over document lengths, and deviation from such a distribution may indicate the presence of artificial documents or near duplicates.[5] We compute the character length distribution and present results for three corpora in Figure 4 (additional results in Appendix B.2.3).

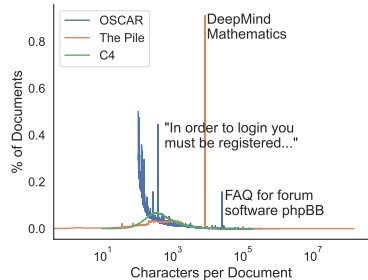

Figure 4: Distribution over character document lengths (in log-scale) for *C4*, *OSCAR* and *The Pile*.

While *C4* is free of duplicate documents, it include clusters of template-generated near-duplicate documents exposed by outliers of identical document lengths. Beyond template-generated user-facing copy (e.g., template-generated documents from a reverse phone lookup site, each associated with a unique phone number), we find clusters of template-generated JavaScript snippets, and large collections of unique documents, including numerous permutations of the same keywords, likely crafted for SEO purposes.

*The Pile*, featuring the longest documents, has a notable outlier with nearly 1% of its documents precisely 8,194 characters long. These outliers are derived from the DeepMind Mathematics dataset (Saxton et al., 2019), truncated to fit this length. *The Pile* also contains a significant number of short template-generated code snippets, e.g., a number of documents (of lengths 9, 18, and 36 tokens) each corresponding to a unique publication in various medical journals, and to auto-generated metadata files (of length 20 tokens) used in the Unity game engine. While *OSCAR* has no documents shorter than 100 characters, as those were filtered, it contains many near-duplicate documents that correspond to website boilerplate, e.g., template-generated FAQs about how to use the forum software phpBB.

### 4.4 Community- and Society-Relevant Measurements

> **Main Findings**
>
> • Instances of popular benchmarks like GLUE and SuperGLUE, were found in various corpora (e.g., *C4* and *RedPajama*), render them unusable for fair model evaluation.
> • Automatic toxicity detection reveals that 1–16.5% of the documents in the corpora contain toxic language using an automatic classifier and between 0.01-16.6% using a taxonomy.
> • An estimated 200M, 4B, and 97M of email addresses, phone numbers, and IP addresses were found in the most PII-contaminated corpora per token (*mC4-en*).

### 4.4.1 Benchmark Contamination

As corpora grow and new evaluation datasets are created, the risk of contamination—where evaluation data are included in a (pre)training corpus—increases. As such, it is important to track contamination (Sainz et al., 2023; Jacovi et al., 2023).[6] Using Search, we provide a contamination analysis of 82 datasets for four popular corpora: *The Pile*, *C4*, *RedPajama*, and *OSCAR*. We consider all datasets

---

[4]Many of these duplicate documents indicate that the user agent used to collect the dataset received automatic responses blocking it from crawling the website's contents.

[5]Outlier lengths are those whose prevalence across the corpus is significantly higher than neighboring lengths.

[6]When evaluating a model trained on an existing corpus, one should exempt contaminated evaluation sets. However, in the case of new corpus construction, practitioners may use WIMBD for decontaminating *the corpus itself* to maintain the evaluation data integrity.

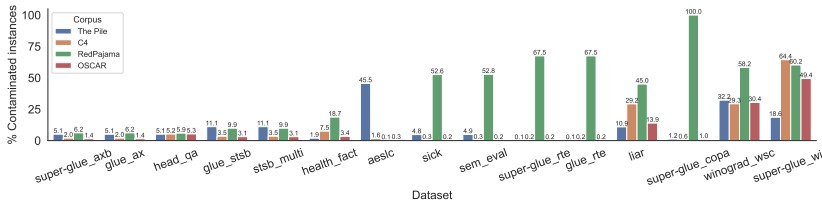

Figure 5: Most contaminated evaluations test sets out of 82 PromptSource (Bach et al., 2022) datasets.

from PromptSource (Bach et al., 2022), a repository containing prompts for 279 different datasets (as of May 2023). We filter datasets we cannot automatically download, from Huggingface datasets (Lhoest et al., 2021), and datasets that do not have a test split. In addition, we only consider datasets that contain at least two inputs (e.g., natural language inference), leaving us with 82 datasets.

We measure contamination by testing whether all input fields are present in a single document and report the percentage of contaminated examples from the test set. Our contamination evaluation serves as an upper bound of exact-match dataset contamination. We provide more details of our analysis and design choices in Appendix B.3.1.

**Contaminated datasets** We present the results in Figure 5. We showcase all benchmarks whose contamination percentages are at least 5% in one of the four corpora. We find that *RedPajama* is the most contaminated dataset out of the four, where in eight out of the 15 corpora, its contamination rate is above 50%, and fully contaminated in the case of COPA (Roemmele et al., 2011). *The Pile*'s contamination rates are lower, but it is also contaminated with a few datasets, such as aesic (Zhang & Tetreault, 2019), WSC (Levesque et al., 2012) and WIC (Pilehvar & Camacho-Collados, 2019), which were included in the SuperGLUE evaluation benchmark (Wang et al., 2019).

**Most examined datasets were not found in the corpora.** It is important to note that while we find some contamination, most of the considered benchmarks do not appear in the corpora we investigated (67 out of the 82 datasets). For instance, Winogrande (Sakaguchi et al., 2021), a large corpus in the style of the Winograd schema, does not appear in any of the examined corpora.

### 4.4.2 PERSONALLY IDENTIFIABLE INFORMATION

PII is "information which can be used to distinguish or trace an individual's identity, such as their name, social security number, biometric records, etc." (Johnson III, 2007). Recent research has sought to *extract* PII from LMs (Carlini et al., 2021). These attacks highlight that LMs can ingest and reproduce PII contained in their training data, and show the risks of training on data that contains such information, even if the data remains private.

We document three kinds of personally identifiable information in pretraining corpora: phone numbers, email addresses, and IP addresses. We employ regular expres-

Table 5: Extrapolated PII frequencies. Count is the extrapolated frequency and *Prec.* is our identification precision accuracy, estimated by manual analysis of 100 random examples.

| Corpus | Email Addresses | | Phone Numbers | | IP Addresses | |
|---|---|---|---|---|---|---|
| | Count | Prec. | Count | Prec. | Count | Prec. |
| OpenWebText | 364K | *99* | 533K | *87* | 70K | *54* |
| OSCAR | 62.8M | *100* | 107M | *91* | 3.2M | *43* |
| C4 | 7.6M | *99* | 19.7M | *92* | 796K | *56* |
| mC4-en | 201M | *92* | 4B | *66* | 97.8M | *44* |
| The Pile | 19.8M | *43* | 38M | *65* | 4M | *48* |
| RedPajama | 35.2M | *100* | 70.2M | *94* | 1.1M | *30* |
| S2ORC | 630K | *100* | 1.4M | *100* | 0K | *0* |
| peS2o | 418K | *97* | 227K | *31* | 0K | *0* |
| LAION-2B-en | 636K | *94* | 1M | *7* | 0K | *0* |
| The Stack | 4.3M | *53* | 45.4M | *9* | 4.4M | *55* |

sions corresponding to each PII type using the Exact Counts . We provide more details about our methodology, the regexes, additional results, and error analyses in Appendix B.3.2. We conduct a manual analysis to estimate the precision of these methods on all corpora. The results of this analysis, as well as the extrapolated frequency of these matches, are presented in Table 5. Our identification method is highly precise (>80% precision) for email addresses on eight out of 10 corpora, and for phone numbers on five of the 10 corpora. Overall, most corpora contain a high volume of PII information, varying in type based on the corpus. For instance, *RedPajama* contain mainly phone numbers (70.2M) and a smaller amount of IP Addresses (1.1M), but S2ORC and *peS2o* contain mainly email addresses (630K and 418K, respectively) and no IP addresses were identified. The most common PII across corpora is phone numbers, followed by email addresses and IP addresses (except for *The Stack*, which has more IP addresses than email addresses: 4.4M vs. 4.3M, and *peS2o*, which has more email addresses than phone numbers). Finally, we observe that *mC4-en* contains the largest amount of PII, also when controlling for the number of tokens (Table 19 in the Appendix).

## 5 DISCUSSION

Data is one of the most poorly understood and studied components in ML research since "everyone wants to do the model work, not the data work" (Sambasivan et al., 2021). Yet, it is one of the most critical factors for successfully training a state-of-the-art language model. While the benefit of increasing model size is evident from the trend of recent years, it is not enough by itself, as the amount and quality of data are crucial (Kaplan et al., 2020).

**Data Curation** With the increasing data needed to train LMs (and other models for other modalities), it remains challenging to curate high-quality datasets. Besides the technical challenges of composing a large-scale dataset and the decisions that go into making it, these decisions and their influence on the final models are costly to assess due to the high computational resources required to train such models. With WIMBD, we hope to ease the decisions that go into crafting large-scale datasets by surfacing patterns and trends about what goes into them and what is left out from different aspects, such as data quality, community and society measurements, etc. Once decisions upon what data is important, and which should be left out of a dataset, practitioners can filter documents or passages that adhere to such decisions. The curation of the Dolma dataset (Soldaini et al., 2024) that happened while developing this work benefited from iterations over the insights from this work, such as the finding of 'noisy' most-common $n$-grams, and bugs in the initial 'de-duplication' implementation.

**Data Documentation** Adding to previous works that call for more data documentation, such as Datasheets (Gebru et al., 2021) and Data Statements (McMillan-Major et al., 2023), we argue for the importance of documenting such information. While previous works often focused and tailored the documentation for supervised-style datasets (e.g., "Is there a label or target associated with each instance?", "How was the data associated with each instance acquired?" from Datasheets, and "What are the demographic characteristics of the annotators and annotation guideline developers?" from Data Statements) we call for more tailored documentation of large-scale pretraining corpora.[7] This work offers a superset of the automatic full-corpus analyses proposed by Dodge et al. (2021); Gao et al. (2020), with several additions, categorization, and programmatic interface, allowing better understanding of the content of current and future large text corpora.

**Grounding Models to their Training Data** Unlike other factors of language model training, such as model architecture or optimizer choice, training data comes in the same natural language format as language model's outputs and thus can be measured and described in all the same ways. As such, the data offers a unique opportunity for grounding models. For instance, a model's ability to recall factual knowledge is derived from its training data (Jiang et al., 2020; Elazar et al., 2021a). On the other hand, models often perform better on frequent occurrences (Razeghi et al., 2022a; McCoy et al., 2023), and on documents similar to models' training data (Longpre et al., 2023). The path to a holistic comprehension of model behavior is through the data, which requires an infrastructure investment to access big datasets and the right abstraction of data attributes.

## 6 CONCLUSION

In this work, we propose WIMBD, a framework for processing and analyzing large text corpora. Using WIMBD, we study ten different corpora that were used to train language models (or vision and language models, such as Stable Diffusion). We uncover interesting insights about these corpora using sixteen different analyses across four aspects: high-level statistics, data quality, community- and society- relevant measurements, and cross-data analysis. For instance, the most common source of texts for the *LAION-2B-en* dataset are the commercial websites Pinterest, Shopify, SlidePlayer, Amazon, and eBay. Regarding data quality, we find that about 50% of *RedPajama* and *LAION-2B-en*'s documents are duplicates. In addition, we find that many evaluation benchmarks, including several from GLUE and SuperGLUE, such as WSC, WIC, and RTE, are contaminated due to their appearance in corpora such as RedPajama. Besides the analyses, WIMBD offers an extendable platform for reproducing our analyses on other corpora, developing new ones, and answering research questions about data. We release all the code and artifacts for WIMBD to encourage researchers to adopt and extend our framework and analyze existing and new corpora.

---

[7]Many questions are still relevant for large pretraining corpora (e.g., "What do the instances that comprise the dataset represent (e.g., documents, photos, people, countries)?").

ACKNOWLEDGMENTS

We want to thank Ludwig Schmidt, Maarten Sap, and Emma Strubell, and the anonymous reviewers for discussions and feedback on this paper, Elizabeth Salesky for the help with Unicode rendering and getting excited about obscure Unicode characters with me, and Carissa Schoenick, Jon Borchardt, and Johann Dahm for assisting with visuals.

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

## A  CORPORA: ELABORATION

We cover ten different corpora, including text-only corpora (e.g., *C4*), captions from image-captioning (*LAION-2B-en*), and code (*The Stack*). A high level description of these corpora using WIMBD is presented in Table 2, and details about the information contained in those corpora are detailed in Table 6.

We analyze all corpora fully, including the different subsets (e.g., *The Pile* is constructed of multiple sources, such as Wikipedia, arXiv, etc.). The only exceptions are *mC4*, and *LAION*, which the original released data consist of non-English texts as well, and we focus on the English subset. Note that while we focus on English text corpora, most of our analyses are not language dependent, and can be easily applied to other languages as well. The only exception is the toxic language analysis (§B.3.3) that relies on an English lexicon and classifier. However, we note that given non-English lexicon and classifier, the analysis can be easily repeated for other languages using our framework.

**OPENWEBTEXT**  is an open-source reproduction[8] (Gokaslan & Cohen, 2019) of the data used to train GPT-2 (Radford et al., 2019). Due to the limited information provided by Radford et al. (2019), and never releasing the data, it is unclear how similar *OpenWebText* is to the original data (WebText), but similar steps to the paper's reports were conducted (such as deduplication, non-English filtering, min-length filtering, etc.).

**C4**  is the dataset used by Raffel et al. (2020) for training T5. The dataset: The Colossal Clean Crawled Corpus (*C4* in short) is based on Common Crawl as a source of text that was scraped from the web. As such, a lot of the data is noisy, and a set of heuristics were employed to clean it up, such as filtering documents by length, obscene/bad words, duplicate texts, non-english, etc. *C4* was not released by Raffel et al. (2020), and instead, it was scraped, cleaned, filtered, and released by Dodge et al. (2021).

**MC4-EN**  is a multilingual version of *C4* that was used to train mT5 (Xue et al., 2021), and later umT5 (Chung et al., 2023). We use the latest version (v.3.1.0) which was used to train umT5, containing documents collected from Common Crawl through August 2022, and in practice the portion of the data that is classified as English. The main difference of *mC4-en* over *C4* is a higher confidence by a language classifier (from 0.7 to 0.96), while also allowing a 0.1% random set of documents that contain "bad words" to pass through, and adaptation of the "bad words" list that resulted in filtering more than 10% of the documents in a language.

**OSCAR**  is a multilingual corpus based on Common Crawl (Abadji et al., 2022). It contains a length filter for improving data quality that filters out documents with short sentences. They also annotate the data with different labels, such as the language of the document, adult content, and language identification, which they use for different analyses. It is an ongoing effort, and the corpus is maintained and updated regularly.

**THE PILE**  is a corpus consisting of 22 different domains (Gao et al., 2020). Unlike *C4*, the data was not scrapped from the web and then filtered, but pre-selected, with the motivation that this way the data will be of higher quality. The included domains in *The Pile* are diverse: they include data such as Wikipedia, Github, Arxiv, EuroParl, and more. By design, most datasets are upsampled in the hope to increase data quality, from 1.5x with domains such as OpenSubtitles, up to 3x with Wikipedia. Models such as GPT-J (Wang & Komatsuzaki, 2021), GPT-neo (Black et al., 2022) and Pythia (Biderman et al., 2023) were trained on this dataset.

**REDPAJAMA**  is an open-source version reproduction of the data used to train LLaMA (Touvron et al., 2023), and was used to train RedPajama-INCITE (Together Computer, 2023).

**S2ORC**  is a large corpus of English academic papers, which consists the abstracts, full text, including figures, tables, and references (Lo et al., 2020). The texts are automatically extracted from pdfs and LaTeX sources.

---

[8]skylion007.github.io/OpenWebTextCorpus

**PES2O**   is a derivative of *S2ORC*, cleaned and filtered to obtain a more usable version of the data intended to train language models. We use *peS2o* V2 (Soldaini & Lo, 2023).

**LAION**   is a large dataset of images and captions scraped from Common Crawl (Schuhmann et al., 2022). The main dataset (LAION-5B) contains 5.8 billion examples, of which 2.32 billion of the captions are in English (*LAION-2B-en*), which we use in this work. We focus on the text captions but demonstrate qualitative examples using the associated URLs and images when appropriate.

**THE STACK**   (Kocetkov et al., 2023) is a source-code dataset that was collected for training language models, and parts of it were used to train SantaCoder (Allal et al., 2023) and MPT (Team, 2023). It was compiled from GHArchive[9] with some filters: files that cannot contribute to training code such as binary files, files larger than 1MB, and some extensions. In addition, only repositories with permissive licenses were included (18 license types in the version v1.0, and 193 in version v1.1), and we use the v1.2. While the main purpose of code is to provide machine instructions to perform different functionalities, it also contain natural language in the form of comments: "Roughly 40 natural languages are present in docstrings and comments with English being the most prevalent. In python files, it makes up  96% of the dataset."

Table 6: Metadata information contained in the ten corpora we consider. *Text* refers to the main information contained in those datasets, while the type of text is different, e.g. The Stack contains source code, and LAION2B-en descibes images. *URL* indicates the URL that the document was collected from, or in the case of LAION2B-en, the link to the image that the text refers to. *Scrape Date* is the date that the document was scraped from the web, *Date Added* is the date the data was incorporated into the corpora. *Domain/Lang* indicates a subcategory of the text (e.g. field of study, the source from The Pile, code language in The Stack). *ID* is the document ID. *Has Split* signifies whether or not the released data contains a train-test split.

| Corpus | Text | Url | Scrape Date | Date Added | Domain/Lang | ID | Has Split |
|---|---|---|---|---|---|---|---|
| OpenWebText | ✓ | ✗ | ✗ | ✗ | ✗ | ✓ | ✗ |
| C4 | ✓ | ✓ | ✓ | ✗ | ✗ | ✗ | ✓ |
| mC4-en | ✓ | ✓ | ✓ | ✓ | ✓ | ✓ | ✓ |
| OSCAR | ✓ | ✓ | ✓ | ✗ | ✓ | ✓ | ✗ |
| The Pile | ✓ | ✗ | ✗ | ✗ | ✓ | ✗ | ✓ |
| RedPajama | ✓ | ✓ | ✓ | ✓ | ✓ | ✓ | ✗ |
| S2ORC | ✓ | ✗ | ✓ | ✓ | ✓ | ✓ | ✗ |
| peS2o | ✓ | ✗ | ✓ | ✓ | ✓ | ✓ | ✓ |
| LAION-2B-en | ✓ | ✓ | ✗ | ✗ | ✗ | ✓ | ✗ |
| The Stack | ✓ | ✗ | ✓ | ✓ | ✓ | ✓ | ✗ |

---

[9]https://gharchive.org/

| Corpus | 1 | 25 | 50 | 75 | 99 | $N.$ |
|---|---|---|---|---|---|---|
| C4 | 26 | 264 | 964 | 3,886 | 137,117 | 15,668,300 |
| OSCAR | 21 | 303 | 1,351 | 6,108 | 440,577 | 15,424,393 |
| LAION-2B-en | 1 | 6 | 11 | 25 | 892 | 1,470,243 |
| mC4-en | 48 | 580 | 1,448 | 5,984 | 477,951 | 62,209,454 |
| RedPajama | 26 | 264 | 963 | 3,882 | 136,937 | 15,658,463 |

Table 7: Internet domain quantiles of each corpora with URL information. The values correspond to the number of tokens from each internet domain quantile. $N.$ corresponds to the number of unique internet domains.

## B  ADDITIONAL RESULTS

We provide additional details and extended results on all the corpora considered in this work. This appendix is structured in a similar way to the structure in the main paper, categorized by the four different high-level analyses: (1) Data Statistics (Appendix B.1), (2) Data Quality (Appendix B.2), (3) Community- and Society-Relevant Measurements (Appendix B.3), and (4) Cross-Data Analysis (Appendix B.4).

### B.1  DATA STATISTICS

The summary statistics are composed of different analyses that mainly involve the additional metadata associated with the textual documents, such as the URL from which the document was extracted, the date it was collected, etc. We also consider some raw statistics about the corpora, described in the main paper (4.2). The analyses we propose for data statistics are the following:

1. Summary statistics (§4.2)
2. Internet domain distribution (§4.2.2, §B.1.1)
3. Internet domain schemes (§B.1.2)
4. Internet domain suffixes (§B.1.3)
5. Utterance date statistics (§B.1.4)
6. Geolocation (§B.1.5)
7. Language distribution (§B.1.6)

### B.1.1  INTERNET DOMAIN DISTRIBUTION

Here, we provide complete analyses on the five corpora that contain URL information in the corpus metadata. Using the *Exact Counts* , we conduct two analyses: (1) each domain is counted per document (yielding documents per domain), and another where each domain is counted per token in the document (yielding tokens per domain). The results are presented in Figure 6, where the (1) document per domain figures are presented on the left, and the (2) document per token figures are presented on the right.

In Table 7, we analyze the number of tokens in each domain, and calculate the 1, 25, 50, 75, and 99 quantiles of these distributions. Interestingly, the 1% quantile in *LAION-2B-en* include domains which have 1-or-less tokens.

### B.1.2  INTERNET DOMAIN SCHEMES

This analysis computes the domain schemes of the associated URLs using the *Exact Counts* . The results are presented in Figure 7. HTTP and HTTPS are two internet protocols, with the latter being an extension of the first that provides more secure communication. While the exact portion of websites across the web that uses each protocol is hard to assess, traffic that goes through Google primarily uses HTTPS - 95%.[10]

---

[10]https://transparencyreport.google.com/https/overview, as of September 16th, 2023.

The trend of recent years shows an increase in the portion of HTTPS-supported websites, and as such, we can use this portion as a proxy for the internet age of a website: HTTP websites are more likely to be older. In addition, the portion of a corpus is an interesting comparison with the reported portion from Google's traffic.

All corpora containing URL information show significant proportions from Google's reports of 95% for the HTTPS protocol. OSCAR contains the highest proportion with 87.6% HTTPS URLs, while C4 is the lowest with only 62.5%.

### B.1.3 INTERNET DOMAIN SUFFIXES

Next, we compute the suffix distribution of the different corpora using the *Exact Counts* and present the results of the ten most common ones in Figure 8. Compared to the internet domain distribution, the suffixes provide us with a higher-level description of the sources of the documents.

Perhaps not surprisingly, the most common suffix is *com*, which is between 60.1% of the documents in *OSCAR* and 77.5% in *LAION-2B-en*. The distribution of suffixes for each dataset exhibits a long tail with a total of over 3,000 different suffixes in the different corpora. While the top 10 typically represent suffixes from English-speaking countries (e.g., *co.uk*, and *ca*), *LAION-2B-en*'s top-10 contains a lot of non-English speaking countries as well, such as Germany (*de*, 0.7%), Russia (*ru*, 0.5%), France (*fr*, 0.4%) and Italy (*it*, 0.4%).

### B.1.4 UTTERANCE DATE STATISTICS

In this section, we examine the temporal diversity of documents from corpora with either reliable creation timestamps in their metadata or URL source information from which creation time can be estimated. Language usage drifts, new concepts are introduced over time, and the truth of much commonsense knowledge depends on the date an utterance was made. While some datasets we consider (*S2ORC* and *peS2o*) have reliable, API-generated creation timestamps, most have creation dates that reflect the time of a document ingestion into the source dataset and not its origin date (*C4*, *mC4-en*, *RedPajama*, and *LAION-2B-en*). To characterize their temporal distribution, we directly count and bin documents by year for those with reliable creation time metadata. For datasets without this information, we fall back on using either the *earliest* date the URL associated with a document was indexed by the Internet Archive or the date of ingestion into the dataset (whichever is earlier).[11] Note that such a procedure does not provide us with the timestamp of the document that was scraped, and as such, it serves as a lower bound on the document's time creation. Given the limitations of the Internet Archive's API, we do this for a 10,000 document random sample of each dataset, which allows a rough estimate of the collection time for documents in these corpora. Results are shown in Figure 9. We can see that *RedPajama* and *OSCAR* are dominated by documents created in the previous five years (as of September 2023), while other datasets have a more substantial proportion of documents from the first half of the 2010s and earlier. Notably, *S2ORC* and pes2o contain a non-negligible fraction of documents from the pre-internet era.

### B.1.5 GEOLOCATION

In this section, we gauge the geographic diversity of corpora with URL source information in their metadata. We use a commercially developed IP database [12] to estimate the country of origin for 100,000 randomly sampled URLs from each of the five corpora with this information included. While there are limitations to using the location of a hosting server as a stand-in for the content creator's location (i.e., websites are not always hosted locally nor in one unique location), it does provide a rough geographic origin for source material. As seen in Figure 10, most web pages across corpora are hosted in the United States, with the bulk of the remainder distributed amongst the anglosphere. This is unsurprising given the focus on English-language sources in the construction of the corpora under consideration.

Table 8: Percentage of documents in English per dataset.

| Corpus | Percentage |
|--------|-----------|
| OpenWebText | 99.68 |
| C4 | 99.67 |
| mC4-en | 99.56 |
| OSCAR | 99.92 |
| The Pile | 96.12 |
| RedPajama | 96.93 |
| S2ORC | 96.44 |
| peS2o | 100.00 |
| LAION-2B-en | 95.90 |

### B.1.6 LANGUAGE DISTRIBUTION

Here, we aim to assess the proportion of languages in all corpora. We use the CLD2[13] classifier to make a prediction about what language is being used in each document, and use this prediction as a label that we analyze in aggregate. Note that we use the classifier label also in mixed-language documents (if CLD2's is_reliable flag is False, we apply the label UN). Table 8 reports the percentages of English-language documents across corpora. As expected, the English fraction is quite high, given the targeted construction of most datasets we consider. The remaining percentages of non-English documents are broken down for the ten remaining most common languages in Figure 11. Note that the classifier we use, as with other classifiers, is imperfect, and as such the identified languages may be wrong.

---

[11]The Internet Archive is a massive library that has been preserving the web since 1996. `https://archive.org`

[12]This work includes IP2Location LITE data available from `https://lite.ip2location.com`

[13]`https://github.com/CLD2Owners/cld2`

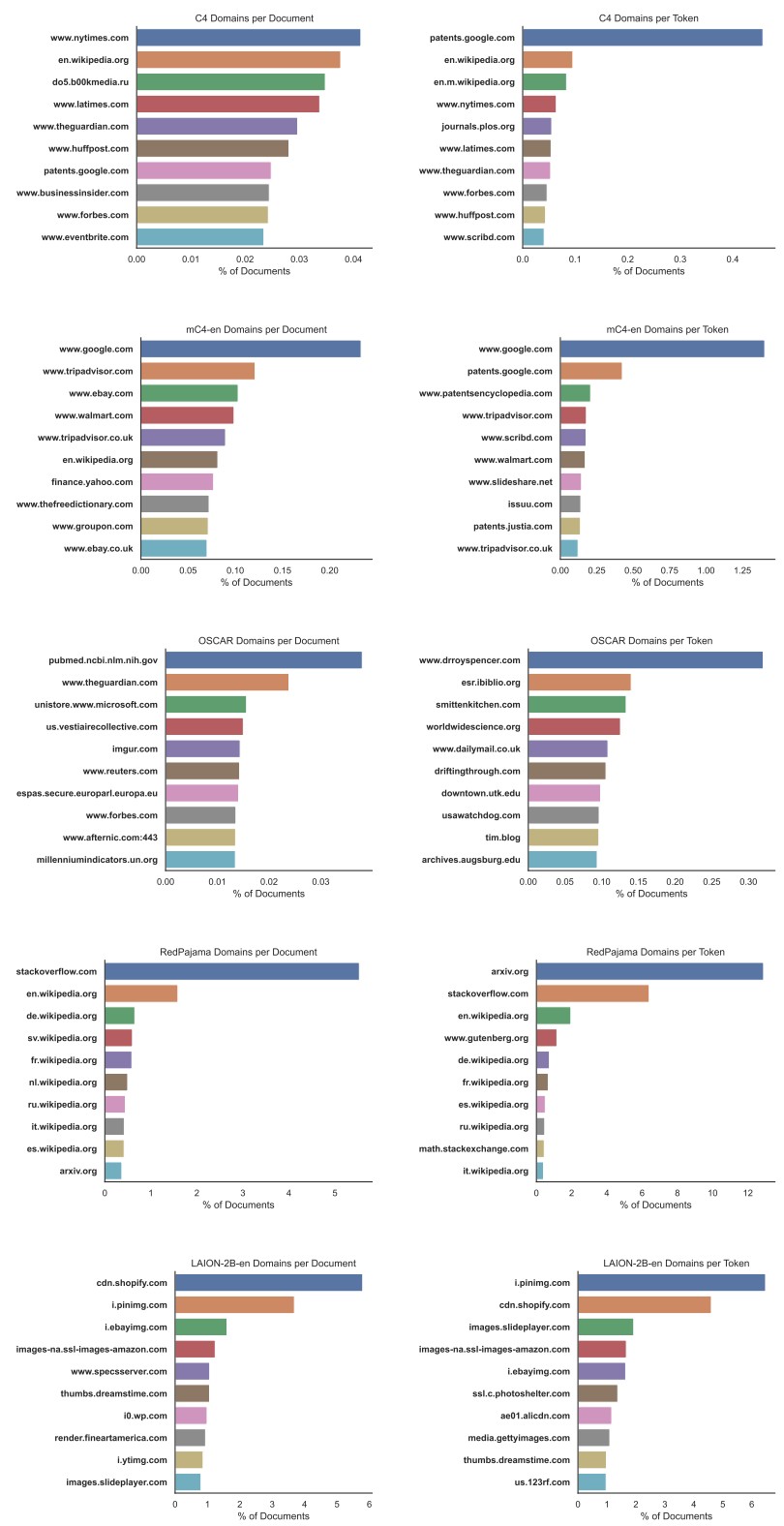

Figure 6: Internet domain distributions of the ten most common domains for each corpus.

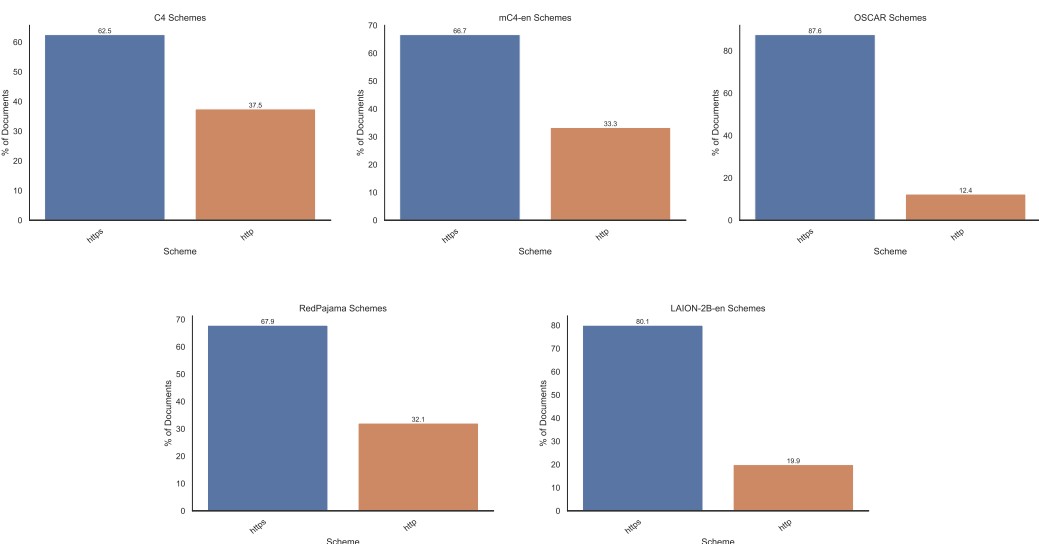

Figure 7: Schema distributions of the ten most common domains for each corpus. We show the results for the five corpora that contain URL information.

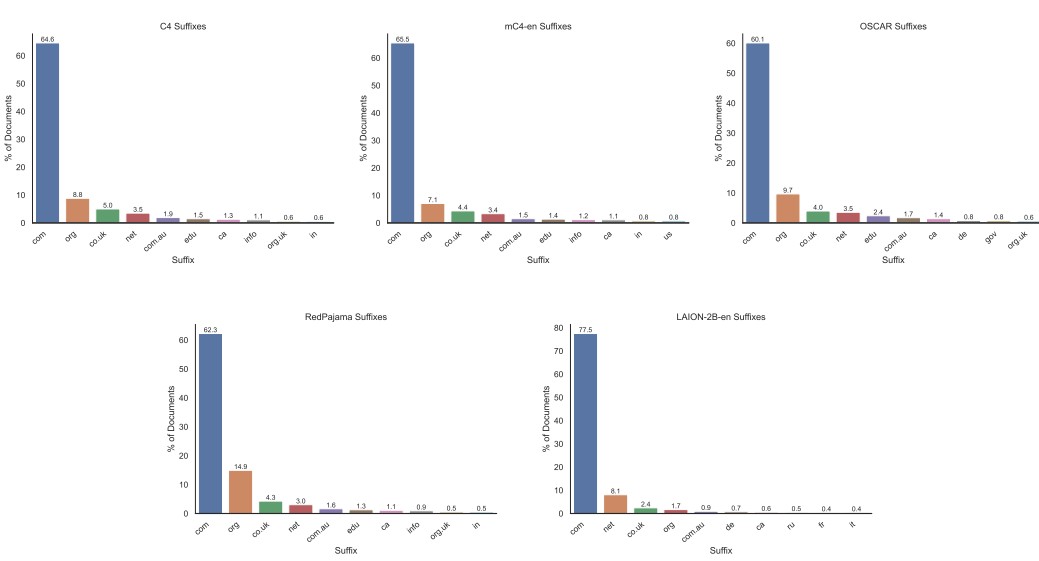

Figure 8: Suffix distributions of the ten most common domains for each corpus. We show the results for the five corpora that contain URL information.

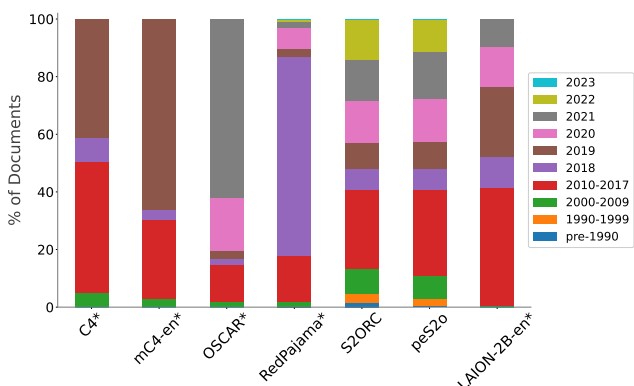

Figure 9: Fraction of documents in each corpus produced per year. Corpora marked with * are estimates based on the Internet Archive index dates for a 10,000 document sample.

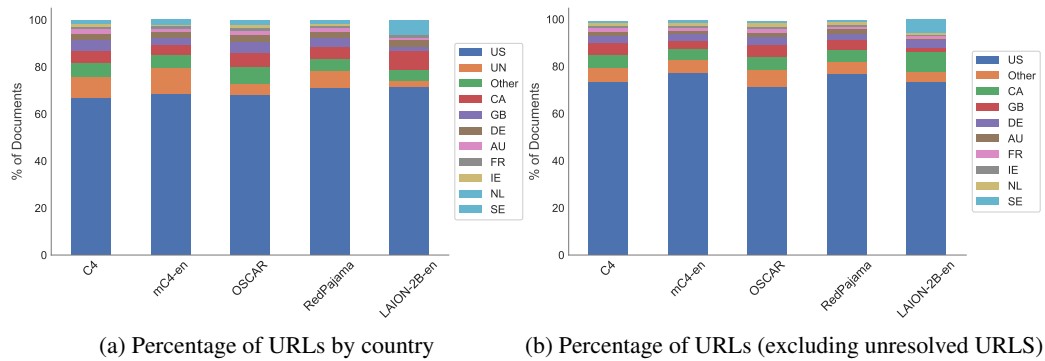

(a) Percentage of URLs by country

(b) Percentage of URLs (excluding unresolved URLS)

Figure 10: Percentage of documents for each dataset originating in a given country. Only the nine most common countries across corpora are shown with the remainder combined in 'other.' We label URLs we were unable to geolocate as UN (Unknown), and provide results with and without these documents included.

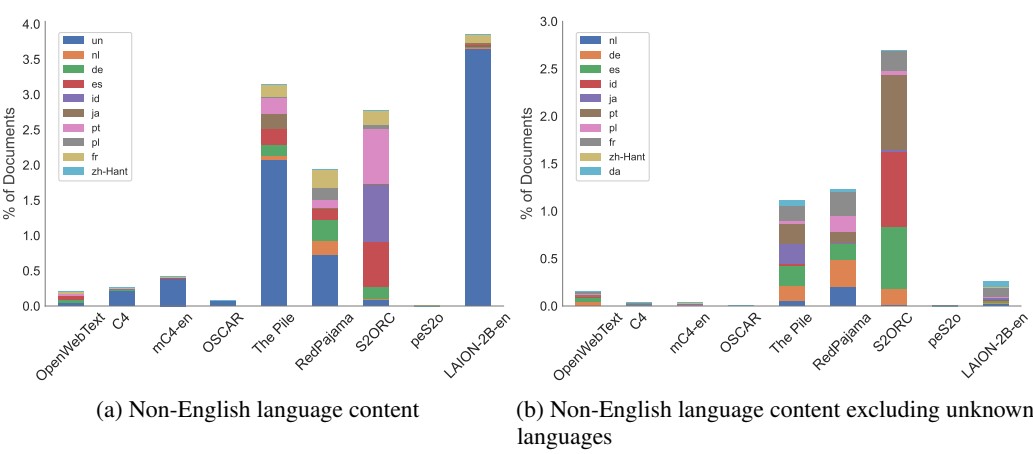

(a) Non-English language content

(b) Non-English language content excluding unknown languages

Figure 11: Percentage of non-English language documents detected in each corpus.

Table 9: Most common unigrams, bigrams and trigrams and their estimated counts.

| OpenWebText | | C4 | | mC4-en | | OSCAR | | The Pile | | RedPajama | | S2ORC | | peS2o | | LAION-2B-en | | The Stack | |
|---|---|---|---|---|---|---|---|---|---|---|---|---|---|---|---|---|---|---|---|
| n-gram | Count | n-gram | Count | n-gram | Count | n-gram | Count | n-gram | Count | n-gram | Count | n-gram | Count | n-gram | Count | n-gram | Count | n-gram | Count |
| **Unigrams** | | | | | | | | | | | | | | | | | | | |
| . | 342M | the | 4.29B | to | 4.29B | to | 4.29B | to | 4.29B | with | 4.29B | the | 2.77B | the | 2.13B | - | 1.13B | } | 4.29B |
| the | 331M | . | 4.29B | the | 4.29B | the | 4.29B | the | 4.29B | to | 4.29B | , | 2.64B | , | 1.9B | , | 870M | { | 4.29B |
| , | 323M | , | 4.29B | of | 4.29B | in | 4.29B | of | 4.29B | the | 4.29B | . | 2.3B | . | 1.69B | the | 578M | the | 4.29B |
| to | 177M | and | 3.87B | and | 4.29B | and | 4.29B | and | 4.29B | that | 4.29B | of | 1.74B | of | 1.35B | " | 455M | class | 4.29B |
| of | 169M | to | 3.67B | a | 4.29B | a | 4.29B | . | 4.29B | on | 4.29B | and | 1.36B | the | 1.05B | class | 352M | a | 4.29B |
| and | 157M | of | 3.29B | , | 4.29B | a | 4.29B | - | 4.29B | of | 4.29B | ) | 1.11B | ) | 769M | of | 341M | \ | 4.29B |
| a | 142M | a | 2.79B | - | 4.29B | . | 4.29B | , | 4.29B | , | 4.29B | ( | 1.11B | ( | 766M | ] | 320M | [ | 4.29B |
| in | 115M | is | 2.17B | , | 4.29B | - | 4.29B | ) | 4.29B | in | 4.29B | , | 1.02B | . | 764M | in | 306M | = | 4.29B |
| - | 91.3M | " | 1.6B | " | 4.29B | " | 4.29B | " | 4.29B | for | 4.29B | in | 985M | . | 749M | / | 249M | > | 4.29B |
| that | 74.9M | - | 1.49B | : | 4.25B | is | 4.26B | ( | 4.28B | as | 4.29B | to | 904M | to | 705M | : | 247M | | |
| **Bigrams** | | | | | | | | | | | | | | | | | | | |
| of the | 39.6M | of the | 740M | of the | 4.29B | of the | 1.85B | - - | 4.29B | of the | 4.29B | of the | 433M | of the | 333M | " " | 257M | ], | 4.29B |
| in the | 29.2M | . The | 608M | in the | 4.29B | , and | 1.5B | of the | 1.3B | , and | 3.65B | . The | 302M | . . | 233M | . . | 96.5M | [ " | 4.29B |
| , and | 29M | , and | 565M | . The | 4.29B | . and | 1.37B | = = | 1.02B | in the | 3.46B | in the | 281M | in the | 208M | of the | 58.2M | class = | 4.29B |
| . The | 27.1M | in the | 523M | . . | 4.29B | in the | 1.28B | . " | 881M | . The | 3.38B | , and | 267M | ). | 206M | in the | 39.5M | ], | 4.29B |
| , the | 19.5M | to the | 321M | , and | 4.29B | . The | 1.17B | , and | 873M | , the | 2.54B | , the | 239M | , and | 181M | T - | 27.8M | > < | 4.29B |
| to the | 16.8M | , the | 296M | , | 4.29B | to the | 825M | * * | 859M | . . | 2.15B | . . | 209M | , the | 162M | for sale | 25.2M | = = | 4.29B |
| . " | 16.5M | ": | 257M | . | 4.29B | . | 4.29B | in the | 805M | . . | 2.06B | ), | 164M | at the | 116M | . . | 22.4M | < / | 4.29B |
| , but | 13.2M | . I | 250M | to the | 4.09B | . I | 704M | . The | 793M | on the | 1.48B | to the | 151M | , and | 111M | , and | 22.4M | ; } | 4.29B |
| on the | 12.8M | for the | 208M | , the | 3.82B | , the | 674M | " " | 774M | and the | 1.32B | ]. | 134M | ]. | 104M | on the | 20.8M | : { | 4.29B |
| . " | 10.9M | . This | 200M | ", | 3.6B | on the | 641M | { \ | 576M | for the | 1.27B | . In | 126M | . In | 97.1M | - Shirt | 19.6M | | |
| **Trigrams** | | | | | | | | | | | | | | | | | | | |
| - - - | 4.67M | . . . | 77.7M | | 4.29B | | 774M | | 4.26B | et al . | 98.6M | et al . | 76.3M | | | " " " | 123M | class = " | 4.29B |
| . . . | 4.6M | . If you | 63.5M | ", " | 2.93B | . . . | 735M | - - - | 686M | al . , | 50.7M | al . , | 38.6M | | | . . . | 49.2M | > < / | 4.29B |
| , and the | 2.46M | . It is | 52.8M | \ \ \ | 2.71B | = = = | 397M | = = = | 926M | - - - | 44.5M | ). The | 34M | | | > < | 19.4M | - - - | 4.29B |
| one of the | 2.42M | as well as | 50.8M | : // | 1.84B | . " " | 473M | : // | 472M | . However . | 35.6M | . However . | 28.3M | | | < br / > | 11.5M | * * * | 4.29B |
| a lot of | 1.74M | one of the | 48.8M | . // | | * * * | 303M | * * * | 326M | q q q | 322M | , and the | 22.5M | | | br / > | 11.5M | " > < | 4.29B |
| . This is | 1.52M | . This is | 43.5M | http : / | 939M | . // | 218M | . . . | 288M | , and the | 311M | . In the | 18.2M | | | for sale in | 10.5M | " : { | 4.29B |
| . It is | 1.51M | , and the | 41.7M | https : / | 832M | . If you | 176M | # # # | 156M | one of the | 287M | ), and | 16.8M | | | : // | 9.58M | " : " | 4.29B |
| , according to | 1.47M | . You can | 38.7M | . , | 675M | ( 1 ) | 152M | ? " " | 133M | . In the | 252M | ( Fig . | 16M | | | Royalty Free Stock | 9.3M | " , " | 4.29B |
| . " The | 1.46M | . However, | 32.3M | . If you | 663M | type = " | 130M | ( 1 ) | 126M | ), and | 23.6M | ]. The | 15.5M | | | http : / | 6.09M | " : " | 4.29B |
| as well as | 1.46M | a lot of | 29.3M | one of the | 619M | . It is | 128M | ] ( # | 117M | ( Fig . | 21.9M | ). In | 14.2M | | | KEEP CALM AND | 5.42M | = = = | 3.98B |
| | | | | | | as well as | 115M | - type = | 116M | https : / | 243M | . . . | 20.8M | | | | | | |

## B.2 DATA QUALITY

While we reported all the different analyses under data quality in the main paper, here we elaborate and provide the full results on all corpora and the different variations (e.g., most common unigrams, bigrams, and length distribution on token level). The analyses we propose for data quality are the following:

1. Most and least common n-grams (§4.3.1, §B.2.1)
2. Duplicate (§4.3.2, §B.2.2)
3. Document length distribution (§4.3.3, §B.2.3)

### B.2.1 MOST & LEAST COMMON n-GRAMS

**Most common n-grams**  In addition to the most common 10-grams reported in Section 4.3.1, we report the results for the most common unigrams, bigrams, and trigrams. Stop words and punctuation are the most common unigrams across the different datasets, with some differences in their ranking. Moving to bigrams, we observe more differences between the corpora. For instance, in *LAION-2B-en*, we observe some marketing mentions, such as "for sale" and "- Shirt". "of the" and "in the" are repeating bigrams in all corpora. In the trigram results, we notice a larger diversion between the corpora. *C4* contains common English expressions, such as "one of the", "a lot of", and "as well as". However, *LAION-2B-en* contains much more marketing material, such as "T - Shirt", "for sale in". *OSCAR* and *The Pile* have many n-grams that look like uncleaned html (": / /", "https : /", "type = "") or markdown ("--", "===", "###").

**Least common n-grams**  Similarly to the most common n-grams, we look at the other side of n-grams distribution on the least common in a corpus. We showcase a random set of 25 unique unigrams from the different corpora in Figures 12 and 13. We observe two noticeable trends from such unigrams: (1) non-standard Unicode fonts like "negative squared latin" (for instance COTD in *mC4-en*), and (2) non-English strings. Non-English strings are quite diverse. The sample from *OpenWebText* contains unigrams from 12 languages other than English: Urdu, Arabic, Korean, Sanskrit, Hebrew, Armenian, Bengali, Persian, Japanese, Latvian, Sindhi, and Russian.

In addition to the unique unigrams inspection, we estimate the number of unique unigrams in each corpus and present the results in Table 10. The unique unigrams results reveal that a non-trivial amount of unique unigrams appear in these corpora. Even the smallest corpus, *OpenWebText*, contains more than 88 million unique unigrams, about 1.1% of the total unigrams in this corpus. The ratio of unique unigrams is about an order of magnitude smaller in the other corpora, except for *LAION-2B-en*, with over 554 million unique unigrams, which constitute 1.8% of the total unigrams.

Table 10: Estimated unique unigrams, and their percentage of the total unigrams.

| Corpus | Count | Percentage |
|---|---:|---:|
| OpenWebText | 88,551,499 | 1.1 |
| C4 | 759,392,762 | 0.5 |
| mC4-en | 4,290,392,741 | 0.2 |
| OSCAR | 1,280,686,454 | 0.3 |
| The Pile | 1,809,241,096 | 0.6 |
| RedPajama | 2,530,085,090 | 0.2 |
| S2ORC | 287,196,445 | 0.5 |
| peS2o | 201,729,350 | 0.5 |
| LAION-2B-en | 554,850,812 | 1.9 |
| The Stack | 4,294,966,820 | 0.3 |

| | | | | |
|---|---|---|---|---|
| مسيحيون | H Y O | 가수들의 | 두분 | بحمد |
| عيادته | ᎤᎦᏘᏄᎤᏗᏛ | 준이에게 | Ḡāzān | شى |
| 라볶이 | পদাবলী | 2 1 2 0 | 미방송영상 | لنضيف |
| त्रिपुरवधार्थमहं | 딱이여라 | وَسَلَامّ | חוֹ | ⍰ |
| שהדברים | دیوانهسى | ゼファル | 시절에도 | создаваемый |

(a) OpenWebText

| | | | | |
|---|---|---|---|---|
| 플래시온은 | *favoured* | 2 B7 | A c c e l e r a t e d | 팔달산에서 |
| 폼일 | 케뮤니케이션 | **nights** | 확실한방향성을 | *BUSINESS* |
| B o p R k | 행위통합 | *added* | I C S | 프로모션버전인 |
| 합니다.Particularly | B G M : john | 학생분들께서는 | 토문 | ⒶⓊⓈⓉⒾⓃ |
| 토폴로지들에 | 평화구조의 | arrived علیه | _ _ _ to | 취발이 |

(b) C4

| | | | | |
|---|---|---|---|---|
| normancomics | 秝 | 🅣🅔🅓🅣🅘🅝🅖 | 🅱🆁🅴🅴🅳 | Tomie |
| forbearance | *pepper* | ✋⍰ | **3980** | ⍰ |
| 🄲🄾🅃🄳 | ξAi | 蛹 | 🄹🄸🄹🄸🄽 | mão |
| δt's | 🄲🄷🄸🄲🄰🄽🄰 | y'all's | HIPSTERS | ⍰ |
| Hostens | coke | 🄱🄸🆁🄳🆂 | 🅂🄷🄰🄽🄽🄰🄷 | *Veggie* |

(c) mC4-en

| | | | | |
|---|---|---|---|---|
| 폭풍구름을 | 2pm | **Sunohara** | *Candy* | 쾌락'이라는 |
| 티벳음악 | 짚곤 | *corniculatus* | الهصحجة | μ0H |
| *Leo* | 홈디제잉 | **1975** | *doll's* | 평택출장안마카톡 |
| 했는OMG | *Franklin* | 한CLSTL녀석 | 최저로 | 🙌 |
| 추산'에 | 통계조사 | e xport | **r**ansi | 준희는B2B |

(d) OSCAR

| | | | | |
|---|---|---|---|---|
| 이윤성 | ⍰ | Bimaبم | N o n o U e | 업데이트하는게 |
| 워크보드시엔 | 사용자들을가져올지 | ⍰ | 털구멍 | ⍰ |
| T r a u r i g | 진흥방안 | ⍰ | ズリ | 이19 |
| 조사받으러 | S P V 2 3 5 | 재생'된 | 슬릿폭에 | ⍰ |
| 시끌쩍하게 | 올라왔기 | 해봐야계군 | *i*20 | 벽전 |

(e) The Pile

Figure 12: Unique unigrams in *OpenWebText*, *C4*, *mC4-en*, *OSCAR*, and *The Pile*.

| | | | | |
|---|---|---|---|---|
| 프루벡 | *ha* | 1 0 . 7 5 2 | 6 2 6 b | 팔하원칙 |
| 폴리부틸렌테레프탈레이트코마와 | 뾊ᇀᄅᄖ⫫ | 확보하게된다.단지 | **los** | |
| 7 4 mm | 햇살청소년사목센터 | *Wherever* | C i p h e r g e n | 프라우다지 |
| 함양출장색시미녀언니 | 7 , 4 5 | 하학이상달 | 토크소로 | MELᴹᴼᵀᴵⱽ |
| 통과시킬때 | 평화주의로 | flageflazione | S Z 1 B | 측정해야만 |

(a) RedPajama

| | | | | |
|---|---|---|---|---|
| 4.22 | فقتلته | 왁스림의 | 자아정체감에 | ἀνατομή |
| ἠστινοσοῦν | يناّله | 학습이론과 | 미백작용 | ساختن |
| 장기운전계획을 | 균들의 | علمانية | 점토의 | Ўj |
| 겨루기 | 작성되어 | ネオレジズム | ▣⌐ | m i c r o e l e c t r o |
| ▣▣▣ | ἐπαίρει | 소유에서 | 찔레꽃 | منينهم |

(b) S2ORC

| | | | | |
|---|---|---|---|---|
| подобрява | filoviridae | ਬਲਿ | êrglis | значительным |
| негативни | OHcomponent | сɪpɪпᴧ | مجره | 튜터 |
| ḥazf | ԡ൱ᖁᑫᖺᑫᏟInč | 혈류량이 | ŻτX | паразитных |
| футуризм | Ḥussein | слабовидящий | бауындай | مدرسين |
| бистатическая | мантия | مصارحة | Ἐπιδέδεικται | Буга |

(c) peS2o

| | | | | |
|---|---|---|---|---|
| ドッチェシリーズフェイン**Hammock** | | 문래창작촌 | 수납박스 | ジャンフィリップ |
| チャイグラス | ウィンドオーケストラ | 푸드스윗 | リトルフジコ | Kennedy |
| 슈퍼주니어_Dancing | ページインタビュー | **BUNDLES** | 알오피 | トップスチュニック |
| フレンズセット | 술이야 | バックローン | クレープデー | 🅐gaaz |
| クーリングマイグレイン | ソカバン | 광고판 | 일반참가자 | fáÑ |

(d) LAION-2B-en

| | | | | |
|---|---|---|---|---|
| ⋀⋂ᛕᴎᴨᴪ | 🏳️‍🌈 | *Bernoulli* | *util* | パ ロヌレホ ポ ルツピ ママロンツヌへ へ |
| ⚽️ | r h e t o r i c | Paloma | 봬β | SACHIN |
| ***DOZEN*** | ૩ɑ̃ | ☝️▣ | 𝒷𝓁𝒶𝓃𝒸 | ΓʌʌɛɪKɪNᚷN |
| ᴄᴊ | *yy*1970 | ੪૭1ᴦᴧ | クリザ゙コ | 🐻○ɩ |
| グ リン | リコウグ ウべ ラギ ンボ | ⊦ ⚒⫤▦ᐁᚠ | ***poppies*** | c z n |

(e) The Stack

Figure 13: Unique unigrams in *RedPajama*, *S2ORC*, *peS2o*, *LAION-2B-en*, and *The Stack*.

Table 11: Top 5 most occurring text duplicates from datasets with duplicates (OpenWebText and C4 don't have any duplicate documents). Truncation for visualization is marked by [...].

| Corpus | Property | #1 Duplicate | #2 Duplicate | #3 Duplicate | #4 Duplicate | #5 Duplicate |
|---|---|---|---|---|---|---|
| mC4-en | Text | ', 'text-align:left; color:white;background-color:#0564d1;'] //}); // ly.show(); var i_type = $("#fa[...] | Tada has the world's leading smart parking technology and has many of the world's top experts. A hug [...] | 4K Ultra-clear picture with exquisite picture quality, plug and play, H.265/H.265+, Max.512G SD card[...] | ', 'text-align:left; color:white;background-color:#0564d1;'] //}); // ly.show(); var i_type = $("#fa[...] | ', marker.on('click', markerClick); if(type==0 && index==0){ marker.emit('click', { target: marker } [...] |
| | Count | 154 | 114 | 80 | 76 | 73 |
| OSCAR | Text | In order to login you must be registered. Registering takes only a few moments but gives you increas[...] | JavaScript is disabled. For a better experience, please enable JavaScript in your browser before pro[...] | Privacy & Cookies: This site uses cookies. By continuing to use this website , you agree to their use[...] | JavaScript seems to be disabled in your browser. For the best experience on our site, be sure to tur[...] | You may not have to, it is up to the administrator of the board as to whether you need to register i[...] |
| | Count | 1,790,064 | 989,919 | 854,143 | 786,678 | 673,136 |
| The Pile | Text | {\n "info" : {\n "version" : 1,\n "author" : "xcode"\n }\n} | \r\n\r\n\r\n  \r\n\r\n\r\n\r\n \tC-Track E-Filing\r\n\t\r\n \t\r\n\t\r\n\t\r\n\r\n\r\n\t\n\t \r\n\t\r\n\t\r\n\t\r\n\t\r\n\t\r\n\t\t\r \n\t\r\n\t\r\n\t\r\n  \r\n\r\n\t\ r\n\t\r\n\t\r\n\t\n[...] | /* Localized versions of Info.plist keys */\n\n | | <?xml version="1.0" encoding="UTF-8"?>\n<!DOCTYPE plist PUBLIC " -//Apple//DTD PLIST 1.0/ /EN" "http://[...] |
| | Count | 3,775 | 2,941 | 2,913 | 2,744 | 2,714 |
| RedPajama | Text | ACCEPTED\n\n#### According to\nInternational Plant Names Index\n\n#### ## Published in\nnull\n\n#### Original n[...] | SYNONYM\n\n#### According to\nThe Catalogue of Life, 3rd January 2011\n\n#### Published in\nnull\n\n#### Ori[...] | ACCEPTED\n\n#### According to\nThe Catalogue of Life, 3rd January 2011\n\n#### Published in\nnull\n\n#### Or[...] | ACCEPTED\n\n#### According to\nNUB Generator [autonym]\n\n#### Published in\nnull\n\n#### Or iginal name\nnull[...] | ACCEPTED\n\n#### According to\nInterim Register of Marine and Nonmarine Genera\n\n#### Published in\nnull\n[...] |
| | Count | 213,922 | 146,434 | 94,922 | 15,038 | 10,089 |
| S2ORC | Text | Abstract not submitted for online publication\n\n\n\n\u2022 Research which is freely available for red istrib[...] | Abstracts P1 - P16 are educational and not included for publication online\n\n\n\n\nO R A L P R E S E N T[...] | Abstract withdrawn\n\n\n \n\u2022 Convenient online submission \n\u2022 Thorough peer review \n\u2022 No space constraints [...] | Educational abstract\n\nO1 Validation of a new automated volumetric breast density measurement system [...] | Modeling and analysis of monkeypox disease using fractional derivatives\n\nThe frequency of monkeypox [...] |
| | Count | 35 | 30 | 26 | 14 | 14 |
| peS2o | Text | Educational abstract\n\nO1 Validation of a new automated volumetric breast density measurement system [...] | Reply on RC2\n\nThis manuscripts investigates the discrepancy of estimated vegetation influence on cat[...] | COP27 climate change conference: urgent action needed for Africa and the world\n\nThe 2022 report of t[...] | Reply on RC2\n\nFollowing your suggestion, we have revised the manuscript very carefully. The lists be[...] | Reply on RC1\n\nThis paper uses a 1D estuary model to explore the variability of overtide under varyin[...] |
| | Count | 14 | 7 | 6 | 4 | 4 |
| LAION-2B-en | Text | Front Cover | Wall View 002 | Market position of the selected technologies | Pointwise: Reliable CFD meshing | Go to European Commission website |
| | Count | 1,003,863 | 681,753 | 414,986 | 319,524 | 314,423 |
| The Stack | Text | #\n%\nRailCompiler: Invalid movement.\n | //\n// WechatAuthSDK.h\n// WechatAuthSDK\n//\n// Created by \u674e\u51ef on 13-11-29.\n// Copyright (c) 2013\u5e74 T[...] | OUTPUT_FORMAT ("elf32-littlearm", "elf32-bigarm", "elf32-littlearm")\nENTRY(reset_handler)\nSEARCH_DIR[...] | //\n// WBHttpRequest+WeiboToken.h\n// WeiboSDK\n//\n// Created by Dannion Qiu on 14/11/6.\n// Copyrigh[...] | //\n// WXApi.h\n// \u6240\ u6709Api\u63a5\u53e3\n//\n// Created by Wechat on 12-2-28.\n// Copyright (c) 2012\u5e74 Tencent. A ll[...] |
| | Count | 45 | 43 | 29 | 24 | 20 |

Table 12: Top 5 most occurring URL duplicates from datasets with URLs for each document and non-zero URL duplication.

| LAION-2B-en | | OSCAR | |
|---|---|---|---|
| Text | Count | Text | Count |
| UNLIKELY | 33,142 | https://international.thenewslens.com/tag/ | 2,184 |
| http://semantic.gs/driver_download_images/driver_download_certifications.png | 27,162 | https://arc.link/twitch/streaming/ | 235 |
| http://www.slickcar.com/products/hawkpadsa.jpg | 10,700 | https://zakiganj24news.blogspot.com/ | 100 |
| https://www.zeitauktion.info/assets/img/zeitauktion_placeholder.jpg | 10,144 | https://ywttvnews.com | 100 |
| https://static.uk.groupon-content.net/app/00/00/default0000.jpg | 9,935 | https://yellgh.com/our-services/ | 100 |

### B.2.2 DUPLICATES

**URL Duplicates**   We also examine duplication between document URLs for the datasets that have that metadata, which we show the top-5 URL duplicates from datasets with URL duplicates in Table 12. LAION's most frequent URL (with 33,142 occurrences) is an invalid URL – "UNLIKELY", likely resulting from a parsing error. The second most frequent URL (with 27,162 occurrences) from *LAION-2B-en* leads to an all-white image from a computer driver website, and in Figure 15, we see that among the top 25 duplicated URLs in LAION-2B-en, there are instances of image duplicates hosted at different URLs. Meanwhile, *OSCAR* has a notable artifact wherein, after the top two duplicate URLs, the next 234 URLs are duplicated exactly 100 times. Table 14 in the Appendix shows counts and ratios for these URL duplicates as previously specified for text hashes. These find that URL duplicate ratios are roughly an order of magnitude smaller than their text hash counterparts, and that the count of documents duplicated by URL is not dominated by only a few clusters.

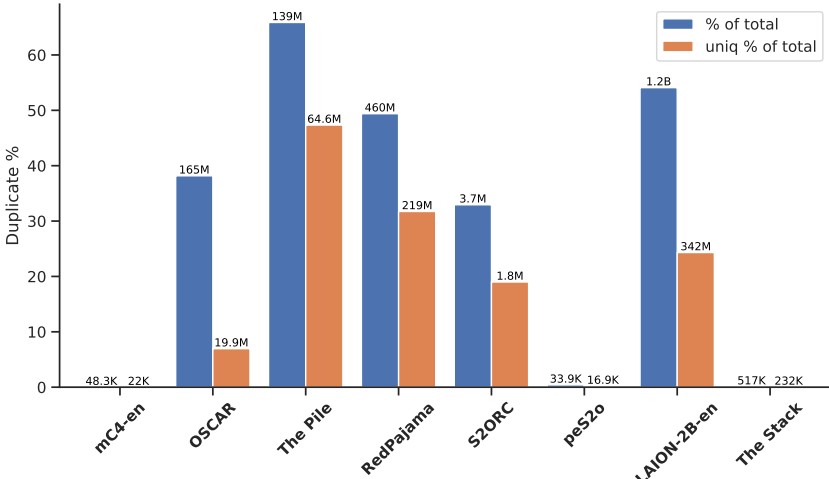

Figure 14: Percentages of text duplicates to totals for datasets with any. The percentages of documents and percentages of unique document clusters are each shown as bars. Duplicate counts are presented above the bars.

Table 13: Statistics about text duplicates per dataset. Counts of duplicate documents and ratio of duplicate to total documents as well as equivalent counts for unique text clusters.

| Corpus | Duplicates | Ratio of total | Unique duplicates | Uniq ratio of total |
|---|---|---|---|---|
| OpenWebText | 0 | 0.00 | 0 | 0.00 |
| C4 | 0 | 0.00 | 0 | 0.00 |
| mC4-en | 48,255 | 0.00 | 21,991 | 0.00 |
| OSCAR | 164,740,386 | 0.38 | 19,934,531 | 0.07 |
| The Pile | 138,716,558 | 0.66 | 64,623,824 | 0.47 |
| RedPajama | 459,530,754 | 0.49 | 218,875,070 | 0.32 |
| S2ORC | 3,703,001 | 0.33 | 1,767,564 | 0.19 |
| peS2o | 33,903 | 0.00 | 16,924 | 0.00 |
| LAION-2B-en | 1,254,910,523 | 0.54 | 342,174,466 | 0.24 |
| The Stack | 517,396 | 0.00 | 232,151 | 0.00 |

### B.2.3 DOCUMENT LENGTH DISTRIBUTION

We elaborate on the results from the main paper and report the length distribution for all corpora, both for the character and token distribution. Figure 16 showcases these distributions, and Table 15 depicts the median token and character length distributions.

*LAION-2B-en*, containing image alt text, has the smallest average document lengths. Beyond the exact duplicates described above, which commonly describe products (especially home appliances), *LAION-2B-en* also contains a significant number of template-generated alt texts paired with maps describing the location of rental boats. The only outlier in *OpenWebText* in terms of document length

Table 14: Statistics about URL duplicates for datasets with URLs for all documents. Counts of duplicate documents and ratio of duplicate to total documents as well as equivalent counts for unique URL clusters.

| Corpus | Duplicates | Ratio of total | Unique duplicates | Unique ratio of total |
|---|---|---|---|---|
| C4 | 0 | 0.00 | 0 | 0.00 |
| mC4-en | 0 | 0.00 | 0 | 0.00 |
| OSCAR | 5,958,969 | 0.01 | 2,542,577 | 0.01 |
| LAION-2B-en | 158,824,858 | 0.07 | 61,674,276 | 0.03 |

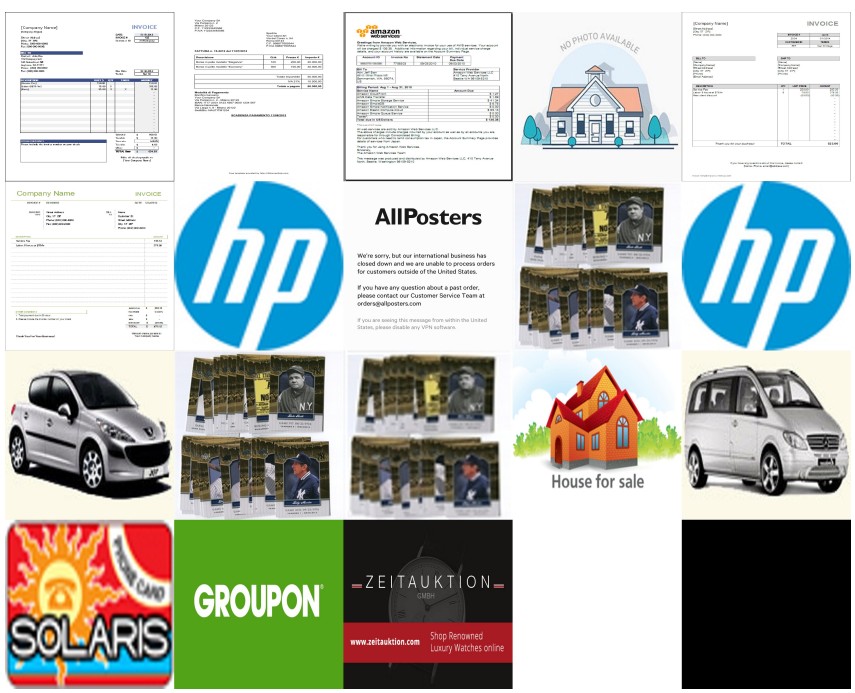

Figure 15: Images from the top 25 most duplicated URLs in *LAION-2B-en*.

is at exactly 100,000 characters; all documents over this length were chunked into multiple documents of length 100,000 by the dataset builders.

*RedPajama* also contains template-generated user-facing copy, including, e.g., placeholder pages for alumni of various secondary schools (each associated with a unique individual's name). This analysis also reveals a collection of documents comprising nearly 0.01% of the dataset, containing what appear to be usernames or titles associated with pornographic content.

Finally, *The Stack* contains many template-generated new-duplicate documents; for example, a large number of auto-generated metadata files for Unity assets, each of length 20 tokens. It also contains a significant number of documents of length 20,000 characters that contain float and bit matrices.

*The Pile* also includes a significant number of auto-generated metadata files corresponding to Unity assets, e.g.:

```
fileFormatVersion: 2
guid: e32f0a7fe2a7abc4289bc3c0e8a2b558
timeCreated: 1435687483
licenseType: Pro
NativeFormatImporter:
userData:
assetBundleName:
assetBundleVariant:
```

as well as auto-generated files corresponding to publications in medical journals, e.g.:

```
{#sp1 .384}
```

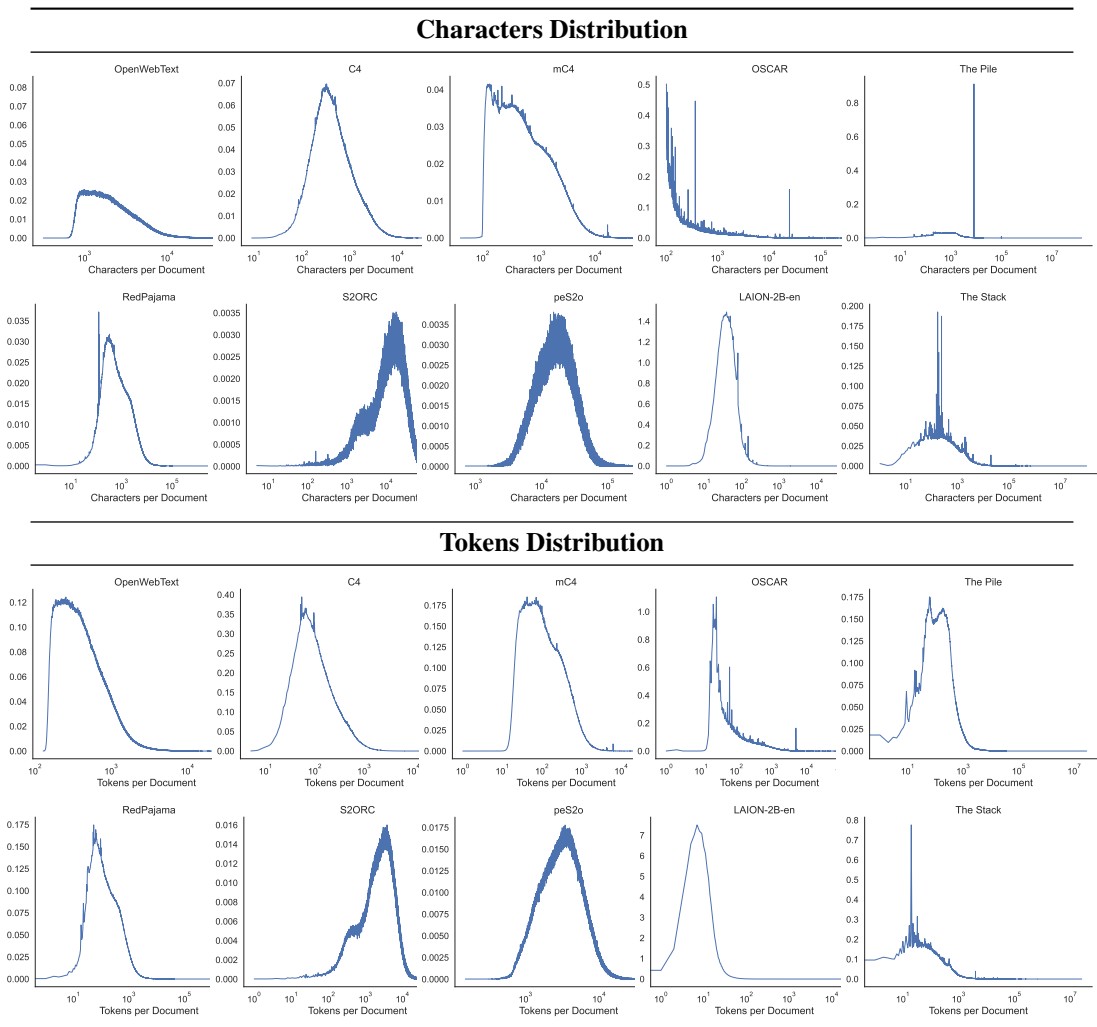

Figure 16: Distribution of document lengths for each of the datasets.

Table 15: Median document lengths for tokens and characters.

| Corpus | Median Token per Document | Median Character per Document |
|---|---|---|
| OpenWebText | 634 | 3,185 |
| C4 | 227 | 1,153 |
| mC4-en | 397 | 1,988 |
| OSCAR | 423 | 2,163 |
| The Pile | 361 | 1,835 |
| RedPajama | 514 | 2,604 |
| S2orc | 4,538 | 23,418 |
| peS2o | 4,582 | 23,852 |
| LAION-2B-en | 10 | 54 |
| The Stack | 430 | 1,953 |

## B.3 COMMUNITY- AND SOCIETY-RELEVANT MEASUREMENTS

In this section, we provide additional results on the contamination and PII analyses from the main paper, as well as conduct two more analyses: toxic language and demographic sentiment co-occurrences. Overall the community- and society-relevant measurements contain the following analyses:

1. Benchmark contamination (§B.3.1)
2. Personally identifiable information (§B.3.2)
3. Toxic language (§B.3.3)
4. Demographic sentiment co-occurrences (§B.3.4)

### B.3.1 BENCHMARK CONTAMINATION

We measure contamination by testing whether all of the input fields are present in a single document, and report the percentage of examples from the test set that are contaminated and present the results in Table 16. We do not test for the presence of the labels as those are not always available, and they can come in different forms (e.g., in RTE they may appear either as 'entailment', 'not-entailment', or as '0', '1'). Moreover, we do not test for consecutive appearance of these inputs, as they might appear in different orders and with different separators. As such, our contamination evaluation serves as an upper bound of exact-match dataset contamination. By employing exact match comparison with the pretraining data, we ignore minor changes in words or phrases that models trained on such similar texts may exploit. An example of such influence is introduced by Emami et al. (2020), who showed how high overlap between sentences in the Winograd Schema Challenge (Levesque et al., 2012) and pretraining corpora inflates the results on the test set, while Elazar et al. (2021b) argue that knowledge and reasoning capabilities from large pretraining corpora leak and inflate evaluation benchmarks.

**Rationales of the Design Choices** Here, we provide the rationals behind our design choices for the contamination experiment. Overall, our desiderata required a large benchmark that can be processed automatically, and that matched in an inspected corpora would be of high precision. We details these rationals in the following points:

- **Choice of task type**. We chose to use tasks that include two or more inputs (e.g., natural language inference) as the co-occurrence of both inputs in the same document increase the likelihood of these inputs to originate from an existing evaluation dataset. In contrary, texts from tasks containing a single input (e.g., sentiment analysis) may naturally occur in some text corpus, which decreases the likelihood of contamination.

- **Ignoring the output**. We decided to ignore the output of the inspected datasets since these can appear in different formats (e.g., numeric values, text labels, etc.).

- **Choice of PromptSource**. Finally, we use PromptSource (Bach et al., 2022) as it is the only large scale benchmark which we could automatically process and discern the different input parts (e.g., this is important since many datasets contain additional fields like metadata which are not directly part of the task).

Note that different design choices can be made for inspecting additional contamination of benchmarks.

Table 16: Contamination percentages of the 82 datasets filtered from PromptSource (Bach et al., 2022), in C4, OSCAR, The Pile, and RedPajama.

| Dataset/Corpus | C4 | OSCAR | The Pile | RedPajama |
|---|---|---|---|---|
| adversarial-qa-adversarialQA | 0.03 | 0.03 | 0.03 | 0.03 |
| adversarial-qa-dbert | 0.00 | 0.00 | 0.00 | 0.00 |
| adversarial-qa-dbidaf | 0.00 | 0.00 | 0.00 | 0.00 |
| adversarial-qa-droberta | 0.10 | 0.10 | 0.10 | 0.10 |
| aeslc | 1.57 | 0.31 | 45.49 | 0.10 |
| amazon-reviews-multi | 2.28 | 2.10 | 1.48 | 2.06 |
| billsum | 0.06 | 0.06 | 0.03 | 0.06 |
| cosmos-qa | 0.00 | 0.00 | 0.00 | 0.00 |
| crows-pairs | 0.00 | 0.20 | 0.00 | 0.60 |
| duorc-ParaphraseRC | 0.00 | 0.00 | 0.00 | 0.00 |
| duorc-SelfRC | 0.01 | 0.00 | 0.02 | 0.02 |
| esnli | 0.04 | 0.08 | 1.13 | 1.24 |
| gigaword | 0.15 | 0.36 | 1.18 | 2.82 |
| glue-ax | 1.99 | 1.45 | 5.07 | 6.16 |
| glue-mnli-matched | 1.65 | 1.77 | 2.17 | 2.26 |
| glue-mnli-mismatched | 1.73 | 1.91 | 2.11 | 2.17 |
| glue-mrpc | 0.06 | 0.00 | 0.64 | 1.16 |
| glue-qnli | 0.13 | 0.04 | 1.48 | 1.21 |
| glue-qnli | 0.09 | 0.04 | 1.48 | 1.21 |
| glue-rte | 0.20 | 0.17 | 0.13 | 67.47 |
| glue-stsb | 3.48 | 3.12 | 11.09 | 9.86 |
| glue-wnli | 0.00 | 0.00 | 0.00 | 2.05 |
| head-qa-en | 5.22 | 5.29 | 5.11 | 5.94 |
| health-fact | 7.53 | 3.40 | 1.94 | 18.70 |
| hlgd | 0.00 | 0.00 | 0.00 | 0.00 |
| liar | 29.23 | 13.95 | 10.91 | 45.05 |
| math-dataset-algebra-linear-1d | 0.00 | 0.00 | 0.00 | 0.00 |
| math-dataset-algebra-linear-2d | 0.00 | 0.00 | 0.00 | 0.00 |
| math-dataset-algebra-linear-2d-composed | 0.00 | 0.00 | 0.00 | 0.00 |
| math-qa | 0.34 | 0.03 | 0.00 | 0.07 |
| mc-taco | 0.00 | 0.00 | 0.00 | 0.14 |
| mocha | 0.00 | 0.00 | 0.00 | 0.03 |
| openai-humaneval | 0.00 | 1.22 | 0.00 | 0.00 |
| paws-x-en | 0.05 | 0.00 | 0.15 | 0.20 |
| paws-labeled-final | 0.05 | 0.04 | 0.25 | 0.35 |
| piqa | 0.06 | 0.03 | 0.06 | 0.13 |
| race-all | 0.14 | 0.06 | 0.00 | 0.28 |
| race-high | 0.11 | 0.00 | 0.00 | 0.26 |
| race-middle | 0.21 | 0.21 | 0.00 | 0.35 |
| ropes | 0.00 | 0.00 | 0.00 | 0.00 |
| samsum | 0.00 | 0.00 | 0.00 | 0.12 |
| scan-addprim-jump | 0.00 | 0.00 | 0.05 | 0.16 |
| scan-addprim-turn | 0.00 | 0.00 | 0.08 | 0.00 |
| scan-filler-num0 | 0.00 | 0.00 | 0.00 | 0.09 |
| scan-length | 0.00 | 0.00 | 0.03 | 0.00 |
| scan-simple | 0.02 | 0.00 | 0.10 | 0.26 |
| scan-template-around | 0.00 | 0.00 | 0.00 | 0.18 |
| scan-template-jump | 0.00 | 0.00 | 0.00 | 0.09 |
| scan-template-opposite | 0.00 | 0.00 | 0.04 | 0.16 |
| scan-template-right | 0.00 | 0.00 | 0.11 | 0.16 |
| scicite | 1.78 | 1.51 | 0.86 | 1.72 |
| scitail-snli-format | 0.09 | 0.38 | 0.28 | 0.71 |
| scitail-tsv-format | 0.09 | 0.38 | 0.28 | 0.71 |
| sem-eval-2014 | 0.35 | 0.18 | 4.89 | 52.81 |
| sick | 0.31 | 0.18 | 4.79 | 52.61 |
| snli | 0.04 | 0.08 | 1.11 | 1.22 |
| squadshifts-amazon | 0.00 | 0.00 | 0.00 | 0.00 |
| squadshifts-new-wiki | 0.01 | 0.01 | 0.01 | 0.03 |
| squadshifts-nyt | 0.01 | 0.03 | 0.02 | 0.04 |
| stsb-multi-mt | 3.48 | 3.12 | 11.09 | 9.86 |
| subjqa-books | 0.00 | 0.00 | 0.00 | 0.00 |
| subjqa-grocery | 0.00 | 0.00 | 0.00 | 0.00 |
| subjqa-movies | 0.00 | 0.00 | 0.00 | 0.00 |
| subjqa-restaurants | 0.00 | 0.00 | 0.00 | 0.00 |
| super-glue-axb | 1.99 | 1.45 | 5.07 | 6.16 |
| super-glue-axg | 0.00 | 0.00 | 0.28 | 0.00 |
| super-glue-boolq | 0.00 | 3.05 | 0.00 | 0.03 |
| super-glue-boolq | 0.00 | 3.05 | 0.00 | 0.03 |
| super-glue-cb | 0.00 | 0.00 | 2.00 | 1.60 |
| super-glue-copa | 0.60 | 1.00 | 1.20 | 100.00 |
| super-glue-multirc | 0.00 | 0.00 | 0.00 | 0.00 |
| super-glue-record | 0.00 | 0.00 | 0.00 | 0.00 |
| super-glue-rte | 0.20 | 0.17 | 0.13 | 67.47 |
| super-glue-wic | 64.43 | 49.43 | 18.57 | 60.21 |
| swag-regular | 2.48 | 1.65 | 2.21 | 2.79 |
| tab-fact-tab | 0.00 | 0.00 | 0.00 | 0.00 |
| wiki-qa | 0.24 | 0.18 | 0.19 | 0.91 |
| winograd-wsc-wsc273 | 29.30 | 30.40 | 32.23 | 58.24 |
| winogrande-winogrande-xl | 0.00 | 0.00 | 0.00 | 0.00 |
| xnli-en | 0.12 | 0.24 | 0.36 | 0.44 |
| xsum | 2.13 | 0.13 | 3.30 | 4.28 |
| zest | 0.00 | 0.00 | 0.00 | 0.00 |

### B.3.2 PII

We use three regular expressions inspired by Subramani et al. (2023) to identify email addresses, phone numbers, and IP addresses across pretraining corpora. In addition, we improved the phone numbers regex for better precision. These regexes provide us with a high precision performance (which we manually evaluate) and allows a fast PII identification. We apply postprocessing rules to the resulting matches, to improve the precision of detecting personal information by seeking to eliminate common classes of false positives (such as ISBN numbers that may be flagged as phone numbers). These rules are enumerated in Table 17.

Applying these regular expressions to the ten corpora we study in the paper, Table 20 contains the number of matches of each PII type in each corpus. For faster processing, we filter documents containing a large amount of special characters (such as documents with >50 consecutive ":)" emoticons). We further normalize this statistic, by the number of tokens in each pretraining dataset, in order to estimate the relative proportion of PII in each corpus. These results are in Table 19. We observe that even when controlling for the number of tokens in the different corpora, *mC4-en* has a large amount of personal information compared to the other pretraining corpora.

We manually evaluate the precision of the heuristics. In order to compute this statistic, we sample 100 examples of strings detected as PII (when available), for the three PII types, over the ten pretraining corpora in this study.These results are in Table 18. The nature of this retrieval task makes it challenging to estimate the recall of our method, and more work is needed on the topic. We show the types of examples that may be incorrectly identified as PII by our method in each corpus in Table 21.

Table 17: Regular expressions and postprocessing rules used to identify three PII types (email/ phone numbers/IP addresses).

| PII Type | Regular Expression | Postprocessing Filter |
|---|---|---|
| Email Addresses | [.\s@,?!;:)(]*([^\s@]+@[^\s@,?!;:)(]+?)[.\s@,?!;:)(]?[\s\n\r] | (1) The username cannot be only "(" 
 (2) There must be a "." in the domain |
| Phone Numbers | \s+(?(\d{3}))?[-\. ]*(\d{3})[-. ]?(\d{4}) | (1) 'ISBN', 'DOI', or "#" cannot appear in a context window of 50 characters from the match 
 (2) Cannot contain URL |
| IP Addresses | (?:(?:25[0-5]\|2[0-4][0-9]\|[01]?[0-9][0-9]?)\.){3} (?:25[0-5]\|2[0-4][0-9]\|[01]?[0-9][0-9]?) | (1) 'ISBN', 'DOI', or "#" cannot appear in a context window of 50 characters from the match |

**Assumptions and Limitations:** We make a number of assumptions in doing this analysis, and we describe them below:

- We choose three types of PII: phone numbers, email addresses and IP addresses. These three types of PII have relatively standardized formats (for example, IP addresses are always 32-bit numbers expressed in dotted decimal format), which allows us to construct regular expressions to search for these information types in text. However, the retrieved information types may not correspond to any one individual— for example, government organizations have email addresses and phone numbers.

- Conversely, many types of personally identifiable information are not easily specifiable in the structured format we use for the information types in this study, and as a result we do not identify them in pretraining corpora.

- While many types of information individually may not appear to identify a specific individual, they can be combined with information elsewhere on the internet to form PII. In this work, we only identify a small proportion of potential personal information that is present in pretraining datasets, but further work is needed to analyze the extent to which pretraining corpora include personal information as well as how this information can be sanitized.

- Finally, we do not claim to estimate the risk level or sensitivity of the information types we extract from the pretraining corpus, acknowledging that this is highly context-dependent and personalized.

Table 18: Extrapolated frequency of matches for regex searches of different kinds of PII (email/ phone numbers/IP addresses) in pretraining corpora. This is computed by multiplying the precision of our PII identification module for each pretraining corpus with the number of detections, in order to estimate the number of *true matches*. *Prec.* contain the precision of our identification method, as estimated by manual verification, on each corpora. Precision indicates the proportion of samples detected that we can reasonably infer as accurately matching the PII type. We sample 100,000 documents from each corpora, and analyze 100 samples of each detected PII type when available. * indicates that less than 100 samples for a PII type were found in a corpus, and we report the precision amongst the available PII detections. The number of samples for these corpora/PII type combinations are as follows: LAION-2B-en /Email Addresses (17), LAION-2B-en /IP Addresses (16), PeS2o/Phone Numbers (13), PeS2o /IP Addresses (12), RedPajama/IP Addresses (95), S2ORC / Email Addresses (10), S2ORC / Phone Numbers (1), S2ORC / IP Addresses (0)

| Corpus | Email Addresses | | Phone Numbers | | IP Addresses | |
|---|---|---|---|---|---|---|
| | Count | Prec. | Count | Prec. | Count | Prec. |
| OpenWebText | 363,789.4 | 99 | 532,929.8 | 87 | 70,430.0 | 54 |
| OSCAR | 62,802,224.0 | 100 | 107,163,132.4 | 91 | 3,237,420.6 | 43 |
| C4 | 7,614,759.2 | 99 | 19,702,198.4 | 92 | 796,494.7 | 56 |
| mC4-en | 201,368,945.0 | 92 | 4,067,997,426.2 | 66 | 97,887,510.2 | 44 |
| The Pile | 19,882,348.2 | 43 | 38,019,831.8 | 65 | 4,078,794.7 | 48 |
| RedPajama | 35,217,396.0 | 100 | 70,264,985.9 | 94 | 1,126,129.5 | *30 |
| S2ORC | 630,130.0 | *100 | 1,465,947.0 | *100 | 0.0 | *0 |
| PeS2o | 418,136.9 | 97 | 226,937.5 | *30.8 | 0.0 | *0 |
| LAION-2B-en | 636,252.1 | *94 | 1,029,066.6 | 7 | 0.0 | *0 |
| The Stack | 4,329,620.3 | 53 | 45,473,381.9 | 9 | 4,481,490.7 | 55 |

Table 19: Extrapolated ratios of PII frequency (the number of PII matches multiplied by the estimated precision), normalized by number of tokens in a corpus ($\frac{PII * Precision}{\#Tokens}$).

| PII Type | Email Addresses | Phone Numbers | IP Addresses |
|---|---|---|---|
| OpenWebText | 0.000047 | 0.000069 | 0.000009 |
| OSCAR | 0.000409 | 0.000698 | 0.000021 |
| C4 | 0.000003 | 0.000007 | 0.000000 |
| mC4-en | 0.000423 | 0.008546 | 0.000206 |
| The Pile | 0.000070 | 0.000133 | 0.000014 |
| RedPajama | 0.000034 | 0.000069 | 0.000001 |
| S2ORC | 0.000011 | 0.000024 | 0.000000 |
| PeS2o | 0.000009 | 0.000005 | 0.000000 |
| LAION-2B-en | 0.000021 | 0.000035 | 0.000000 |
| The Stack | 0.000003 | 0.000030 | 0.000003 |

| Corpus | Email Addresses | Phone Numbers | IP Addresses |
|---|---|---|---|
| OpenWebText | 367,464 | 612,563 | 130,426 |
| OSCAR | 62,802,224 | 117,761,684 | 7,528,885 |
| C4 | 7,691,676 | 21,415,433 | 1,422,312 |
| mC4-en | 218,879,288 | 6,163,632,464 | 222,471,614 |
| The Pile | 46,238,019 | 58,492,049 | 8,497,489 |
| RedPajama | 35,217,396 | 74,749,985 | 3,753,765 |
| S2ORC | 630,130 | 1,465,947 | 373,095 |
| peS2o | 431,069 | 736,810 | 239,912 |
| LAION-2B-en | 676,001 | 14,700,951 | 522,005 |
| The Stack | 8,169,095 | 505,259,799 | 8,148,165 |

Table 20: Frequency of matches for regex searches of different kinds of PII in pretraining corpora.

Table 21: Abbreviated examples of incorrect detections by our method, for each PII type, in each pretraining dataset. The exact span that was matched is in red. Offensive content and personal information have been redacted from the presented examples.

| Corpus | Email Addresses | Phone Numbers | IP Addresses |
|---|---|---|---|
| OpenWebText | skremoved) has joined * trayvonmartin sets ban on *!*@n***.*** * trayvonmartin has kicked whitepower from #n**** | ...2017 limitation 99 pcs. article id 472172730 ean 4012138149625 the model was produced in the usual minichamps... | ... [stdout] awy was overriden from notenoughitems 1.6.1.9.jar 2014-03-24 20:25:06 [info] [minecraft-client]... |
| C4 | "you ever googled our email address? try googling "@fmr.com" and "charity" together, and you will get an idea" | on your mortgage. disclaimer - property reference 100103003249. the information displayed about this property | not load file or assembly ´smswrappers, version = 3.0.0.0 |
| mC4-en | smswrappe wrote in messagenews:a30c91p63 cj6vgr...4lfg7ve8@4ax.com... i bought gta iii at a garage sale and it did not | "stat-major-faults": 1213, "stat-total-memory": 3975217152, "stat-swap-in": 0 | s not constitute the consent required by n.j.a.c. 11.5.6.1 (n) for the advertisement of listings exclusively |
| OSCAR | - | ...a getty images) michael jones9 october 2021 21:53 1633812509 andorra vs england player ratings: phil foden shi... | ...latest update software comes with version number 10.0.0.163. currently the update available in the... |
| The Pile | [@eiguren3].[]data-label="table4" | t undefined behavior. for example, i get that b = 2083899728 and d = -552766888. the persistent thing you are | such damage. // according to ecma-262, sections 8.6.2.2 and 8.6.2.3 you're not // allowed to override rea |
| RedPajama | - | watercolor baby bring a book card printable png v 1525458984 - watercolor baby bring a book card printable png | sh wikipedia) 18:54, 15 july 2013 (utc) if i can. 86.146.46.88 john of reading (talk) 06:38, 25 july 2013 (utc) |
| S2Orc | - | - | - |
| PeS2o | 65%@0.00262 | izona institutional review board (approval number 2003521636a002). at baseline, the participants reported thei | - |
| LAION-2B-en | NWA Democrat-Gazette/Michael Woods –03/15/2015– w@NWAMICHAELW... | queen creek 85142 e cherrywood dr - property id: 1311037210 | gods and glory: war for the throne apk 3.8.10.1 |
| The Stack | remirror/ui@0.7.3 | ermine the vision-agent service is running - hsd 15010872669 - add missing heartbeatresponse-timersecs to the | atoaune — have you upgraded to oracle soa suite 12.2.1.1 and can't find the partitions configuration any l |

Table 22: Toxic language percentages based on a taxonomy and a classifier over entire documents in the corpora we consider. Toxic language statistics in the corpora we consider. The document toxicity (the first two columns) reports the percentage of documents that contain at least one mention of toxic language detected by each of the approaches. The classifier is applied separately on each sentence. The fine-grained taxonomy mention (the last three columns) reports the number of toxic mentions overall, and their relative appearance normalized by the number of tokens in each corpus.

| Corpus | % Documents with Detected Toxicity | | Fine-grained Taxonomy Statistics | | |
| --- | --- | --- | --- | --- | --- |
| | Classifier | Taxonomy | Offensive-minority | Offensive-not-minority | Harmless-minority |
| OpenWebText | 16.47 | 13.8 | 149K (1.92e-05) | 3.55M (4.58e-04) | 13.5M (1.74e-03) |
| C4 | 5.75 | 0.01 | 158K (1.03e-06) | 47 (3.06e-10) | 146M (9.51e-04) |
| mC4-en | 6.09 | 0.15 | 31.4M (1.16e-05) | 6.55M (2.42e-06) | 2.85B (1.05e-03) |
| OSCAR | 9.58 | 8.97 | 8.91M (1.87e-05) | 236M (4.95e-04) | 549M (1.15e-03) |
| The Pile | 8.27 | 7.67 | 4.55M (1.59e-05) | 84.7M (2.96e-04) | 238M (8.32e-04) |
| RedPajama | 10.3 | 7.88 | 15.2M (1.49e-05) | 283M (2.76e-04) | 1.43B (1.40e-03) |
| S2ORC | 10.52 | 16.55 | 95.9K (1.60e-06) | 8.02M (1.34e-04) | 33M (5.52e-04) |
| peS2o | 9.56 | 17.0 | 47.8K (1.09e-06) | 5.96M (1.35e-04) | 26.7M (6.07e-04) |
| LAION2B-en | 1.09 | 0.89 | 2.69M (9.09e-05) | 25.4M (8.55e-04) | 182M (6.14e-03) |
| The Stack | 1.16 | 1.85 | 4.63M (3.04e-06) | 84.8M (5.56e-05) | 228M (1.50e-04) |

### B.3.3 TOXIC LANGUAGE

How common is toxic language used in corpora? We employ two complementary methods for computing toxicity. The first is based on the work of (Zhou et al., 2021), who compiled a lexicon of terms (TOXTRIG) into three categories: *possibly offensive minority identity mentions*, *possibly offensive non-identity mentions*, and *non-offensive minority identity mentions*. It is then used by matching these "toxic triggers" over texts. The model-based method uses an SVM classifier trained on a dataset consisting of 200K examples based on Wikipedia and Twitter to identify toxic language.[14] We apply such a classifier on each sentence separately and consider the document toxic in case any sentence is found to be toxic. We present the results in Table 22. *C4* is the least toxic based on the taxonomy: only 0.01% were found to be toxic, which is expected due to the filters used in the curation process of the dataset. On the other hand, the classifier finds more documents to be toxic: 5.75%, which may indicate subtleties that the lexicon used for filtering documents from *C4* did not catch. *OpenWebText* is the most toxic corpus based on the classifier, while PeS2o is the most toxic one based on the taxonomy, perhaps surprisingly, as it is not a web-based corpus.

**Explicit Content Filtering** The only dataset we analyze that explicitly filtered for toxic content (in the form of keyword matching) is C4. Indeed, the matching category from our analysis are the "Offensive-*" categories. Our analysis, that uses a fine-grained lexicon (Zhou et al., 2021), splits this category into "offensive-minority" and "offensive-not-minority". In C4 we only found 47 mentions of the "offensive-not-minority" category, likely due to a difference in filter used to create C4 and our lexicon. In comparison, other datasets that did not employ such filters contain several million references of such phrases. Interestingly, C4 also contains 158K occurrences of the "offensive-minority" category, which were not filtered from the dataset.

### B.3.4 DEMOGRAPHIC SENTIMENT CO-OCCURRENCES

In this section, we turn to detecting biases in the corpora based on demographic factors. We constructed a set of unigrams and bigrams associated with gender (male and female pronouns), religion (the proper names of several major religions), and race (combinations of racial identifiers and words like man, woman, people, etc.). The sentiment of sentences containing these terms was computed using SpacyTextBlob and averaged over a given corpus. The results for all corpora are shown in Figure 17. *The Stack* is excluded from this analysis since the contexts in which these terms appeared were not typically natural language. Overall, we observe a neutral or weakly positive sentiment for sentences in which most of our demographic terms appear, with the exception of those including 'black' being uniformly more negative across all corpora. With minor exceptions we don't observe substantial variation in the sentiment for individual terms among datasets. The weak positivity seen for all sources is in opposition to a related analysis performed in Gao et al. (2020), which measured weak negativity for most terms. It's likely this is due to differences in the way

---

[14]https://github.com/dimitrismistriotis/alt-profanity-check

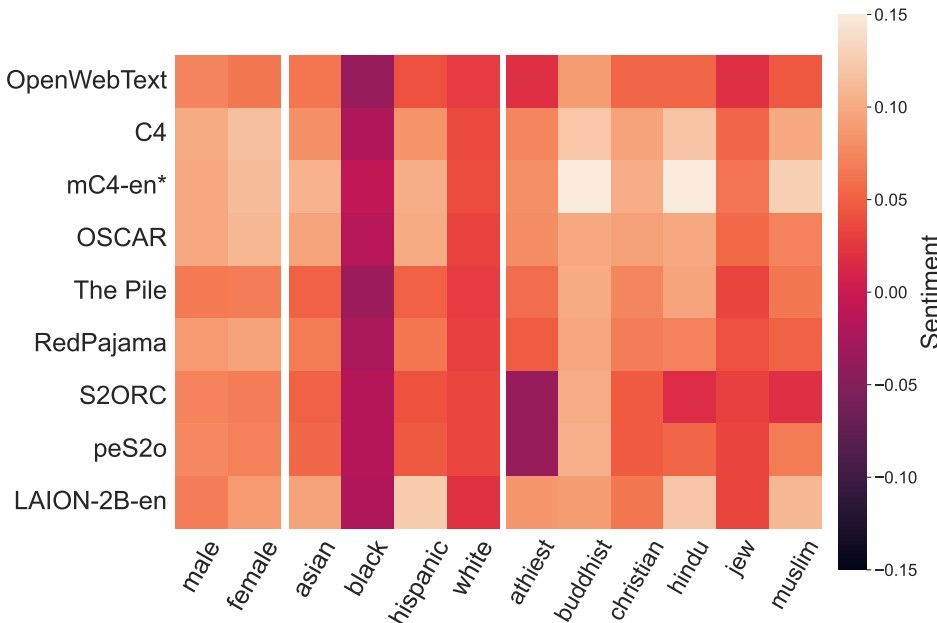

Figure 17: The average sentiment associated with several gender, racial, and religious demographic terms for each dataset. Note: averages for datasets marked with * were computed for 10% samples.

average sentiment is computed (we compute sentiment at the sentence level while Gao et al. (2020) computes sentiment only for the most frequent co-occurring terms).

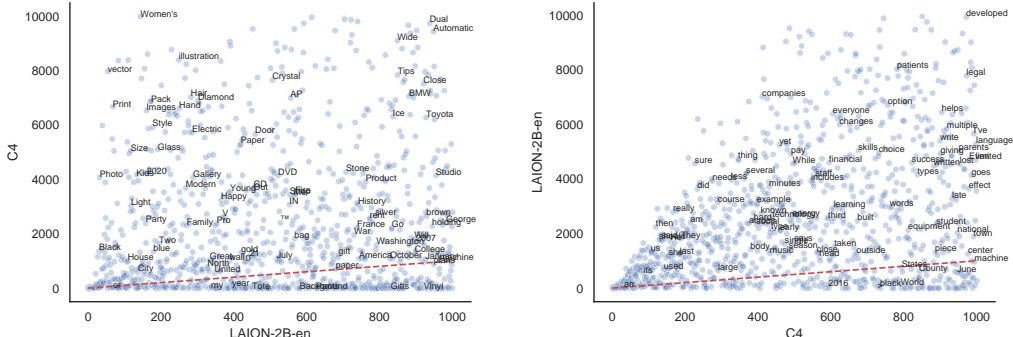

Figure 18: 1,000 most common unigrams in *LAION-2B-en* (rank on $x$-axis), and their corresponding rank in *C4* ($y$-axis), and visa-versa. The dashed red line corresponds to $y = x$. Points below and above that line indicates differences between the corpora. For instance, common unigrams in LAION-2B-en are of different adjectives and words often used to describe objects (e.g., Black, Light, Happy, Woman's), but those are much less common in C4.

## B.4 Cross-Data Analysis

> **Main Findings**
>
> - Comparing unigrams of different corpora reveals distributional and topical differences.
> - *OSCAR* unigram distribution is the most similar to all other corpora on average.
> - 50% of *RedPajama* unique documents originate from *C4* and 50% of *OpenWebText* unique documents originate from *The Pile*.
> - While *mC4-en* was supposedly a superset of *C4*, documents from *C4* constitute only 0.04% of *mC4-en*, while the later being only 10x larger in size.

Using the analyses from the previous sections we can now perform targeted comparisons between different corpora. Such analysis is the first step of better understand the similarities and differences between corpora. We perform the following analyses:

1. Distributional similarities (§B.4.1)
2. Corpus overlap (§B.4.2)

### B.4.1 Distributional Similarity

**Unigram Ranking** Using the most common $n$-gram statistics (4.3.1), we can compare the ranking of these $n$-grams, to gain insights into their different usage between corpora. For the following analysis we consider the top 10,000 most common unigrams of two corpora, and display the 1,000 most common unigrams in one corpus as a function of the same unigram rank in the other corpus. In Figure 18 we display the rank of unigrams in *C4* as a function of their ranks in *LAION-2B-en*. Some very common unigrams in *LAION-2B-en* describing objects such as "Two", "Black", "blue", and "Light" are very common in *LAION-2B-en* - top 500 unigrams, but much more rare in *C4*'s top 1,000. Another category is car models such as BNW and Toyota whose ranking is about 900 in *LAION-2B-en*, but above 6,000 in *C4*. Figures 19-28 show the paired ranks for all corpora pairs.

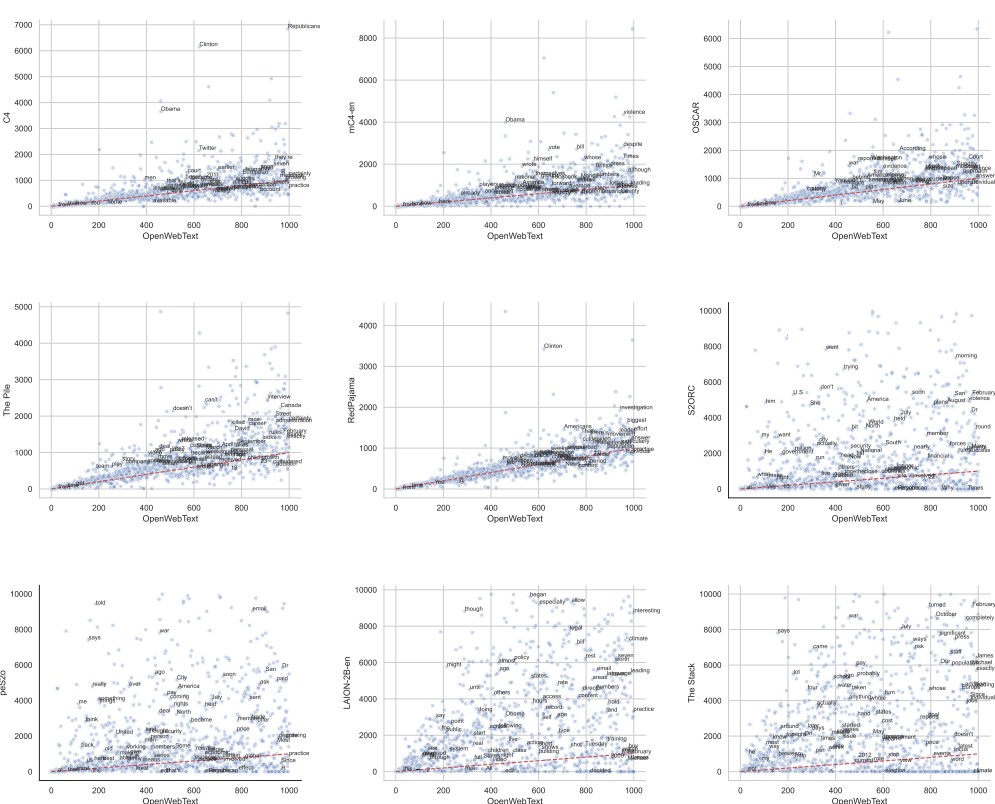

Figure 19: *OpenWebText* top 1,00 unigrams, and their corresponding indices in the other corpora.

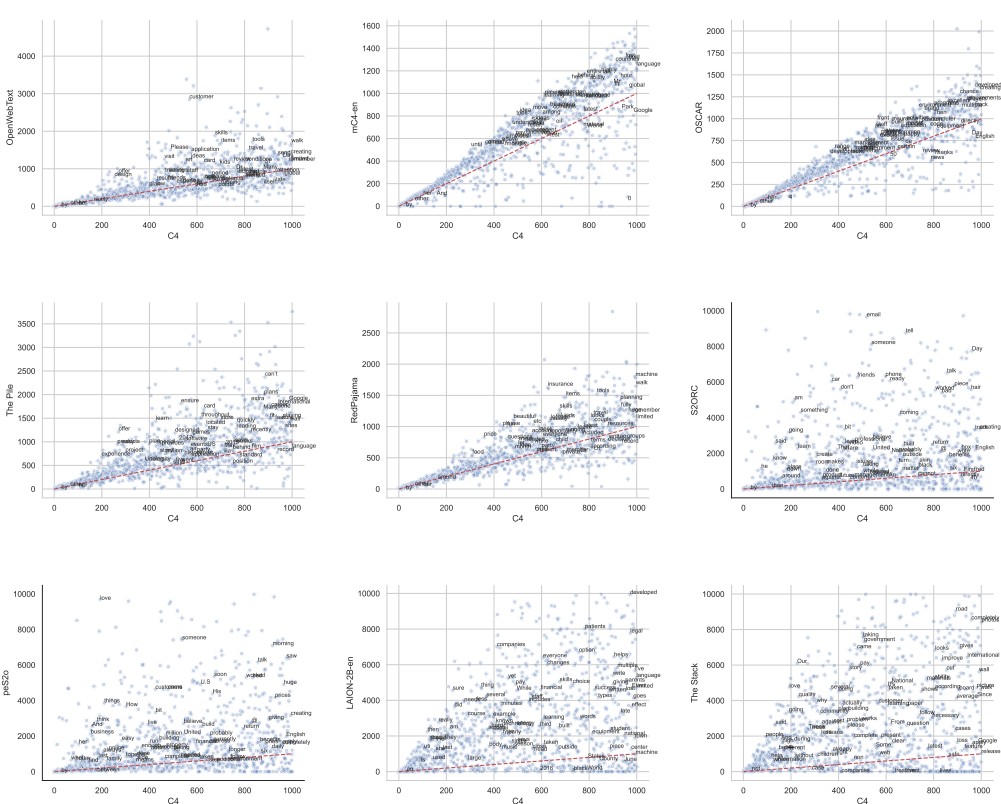

Figure 20: *C4* top 1,00 unigrams, and their corresponding indices in the other corpora.

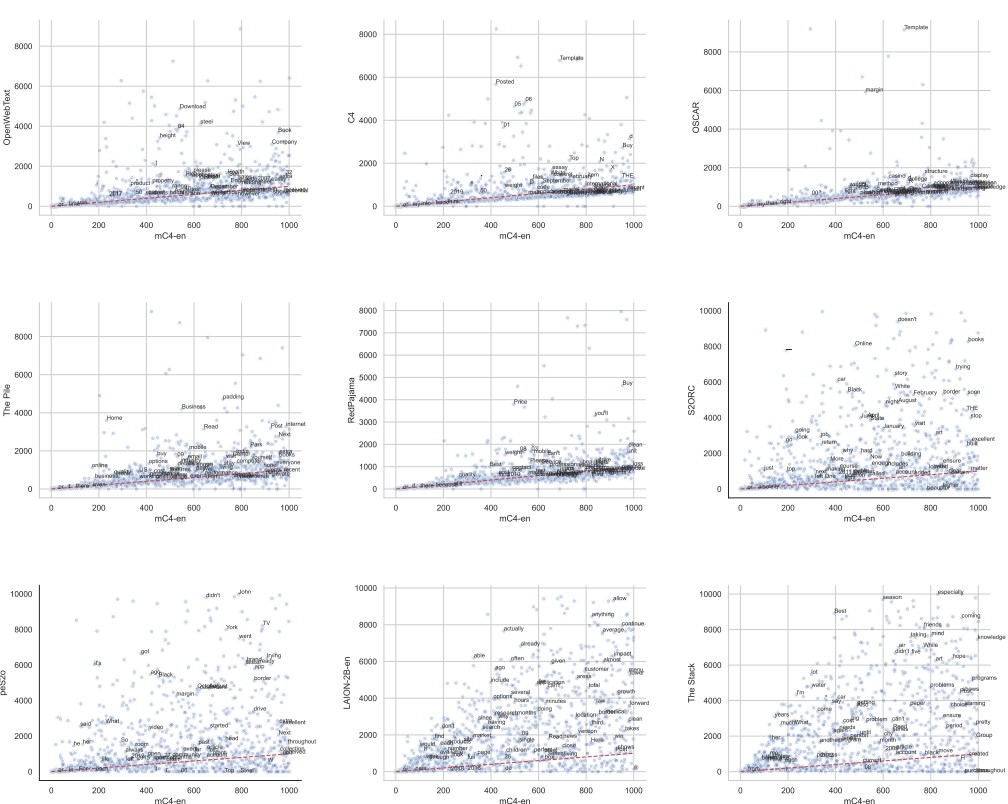

Figure 21: *mC4-en* top 1,00 unigrams, and their corresponding indices in the other corpora.

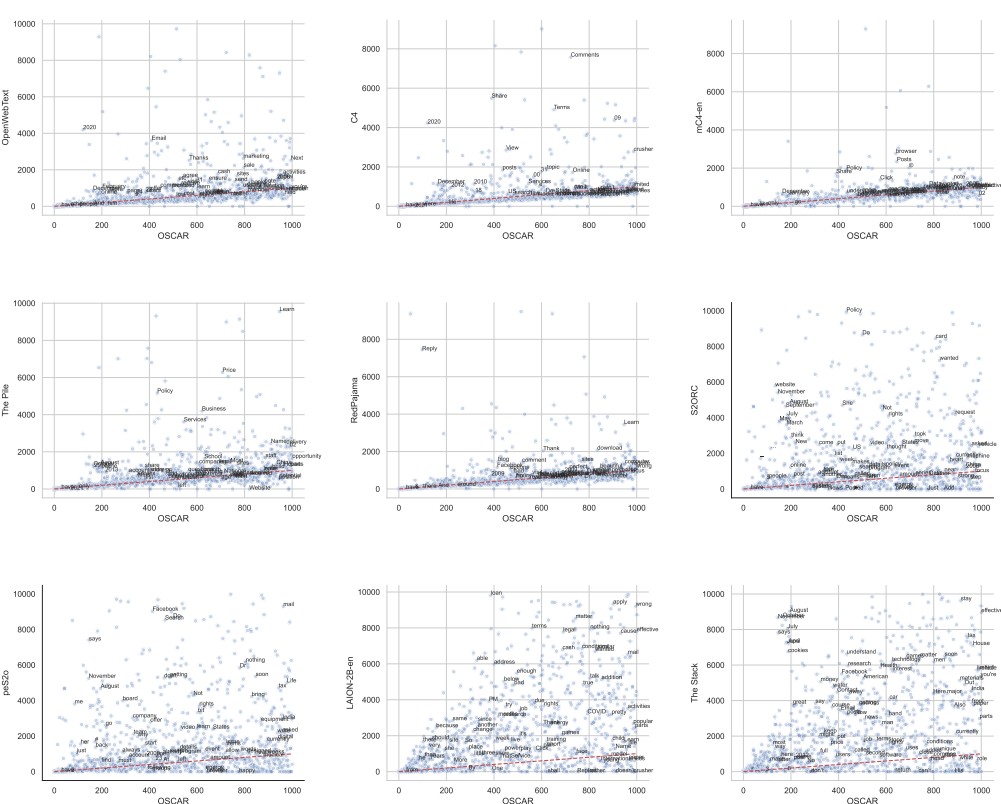

Figure 22: *OSCAR* top 1,00 unigrams, and their corresponding indices in the other corpora.

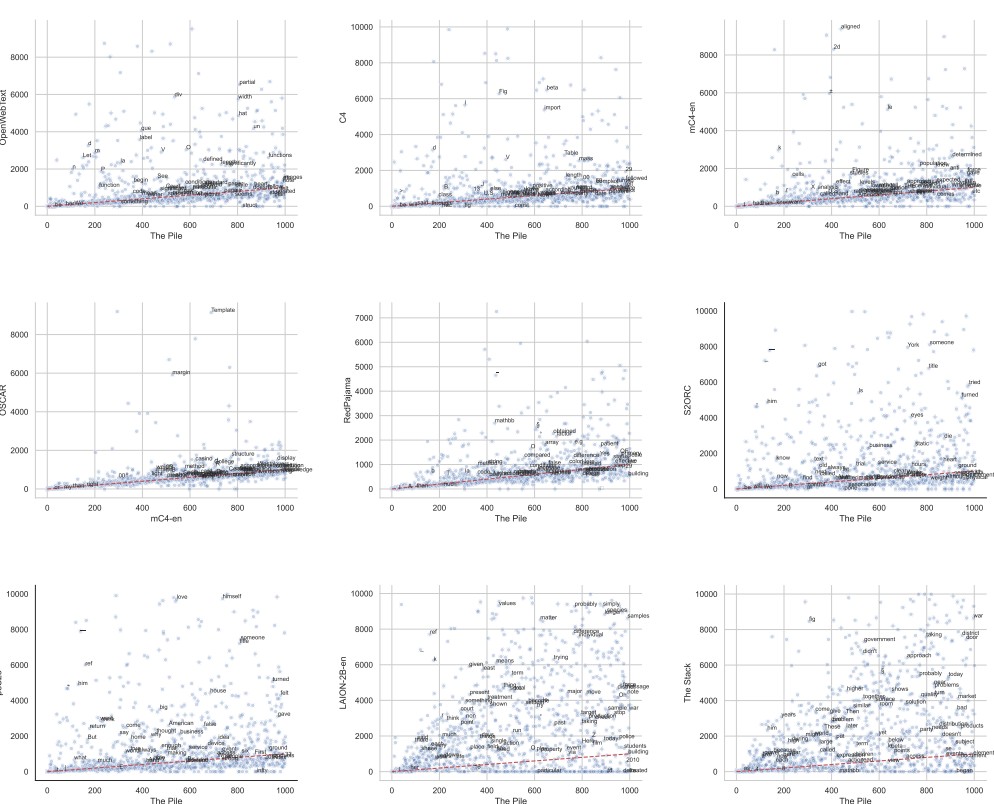

Figure 23: *The Pile* top 1,00 unigrams, and their corresponding indices in the other corpora.

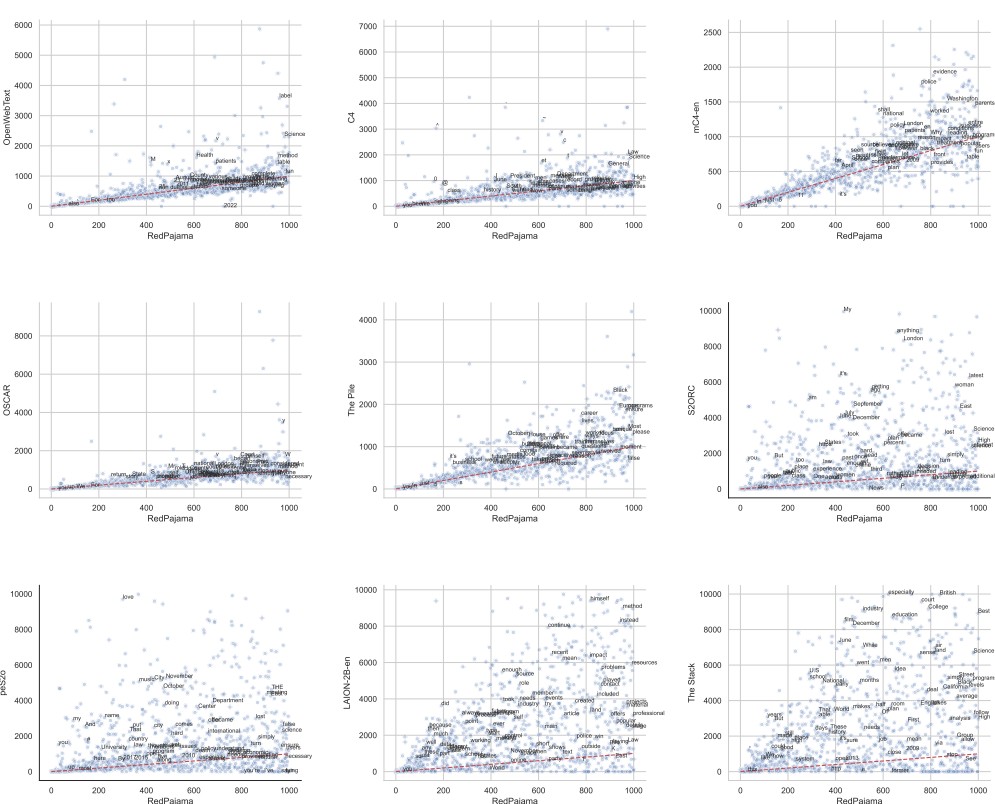

Figure 24: *RedPajama* top 1,00 unigrams, and their corresponding indices in the other corpora.

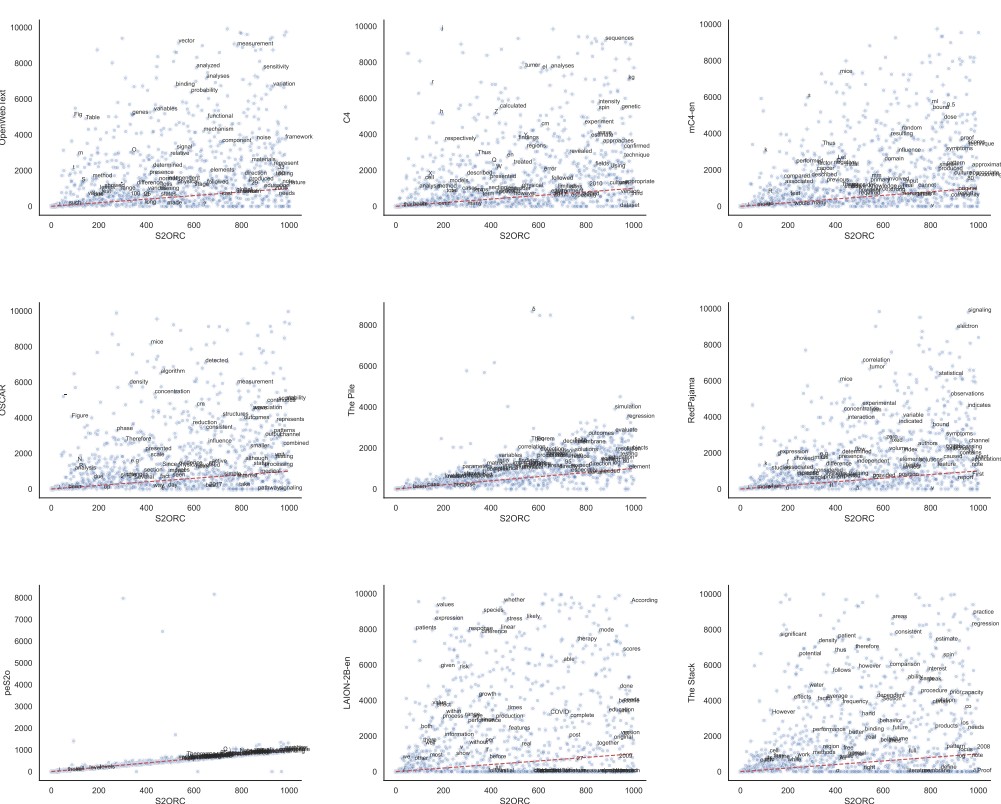

Figure 25: *S2ORC* top 1,00 unigrams, and their corresponding indices in the other corpora.

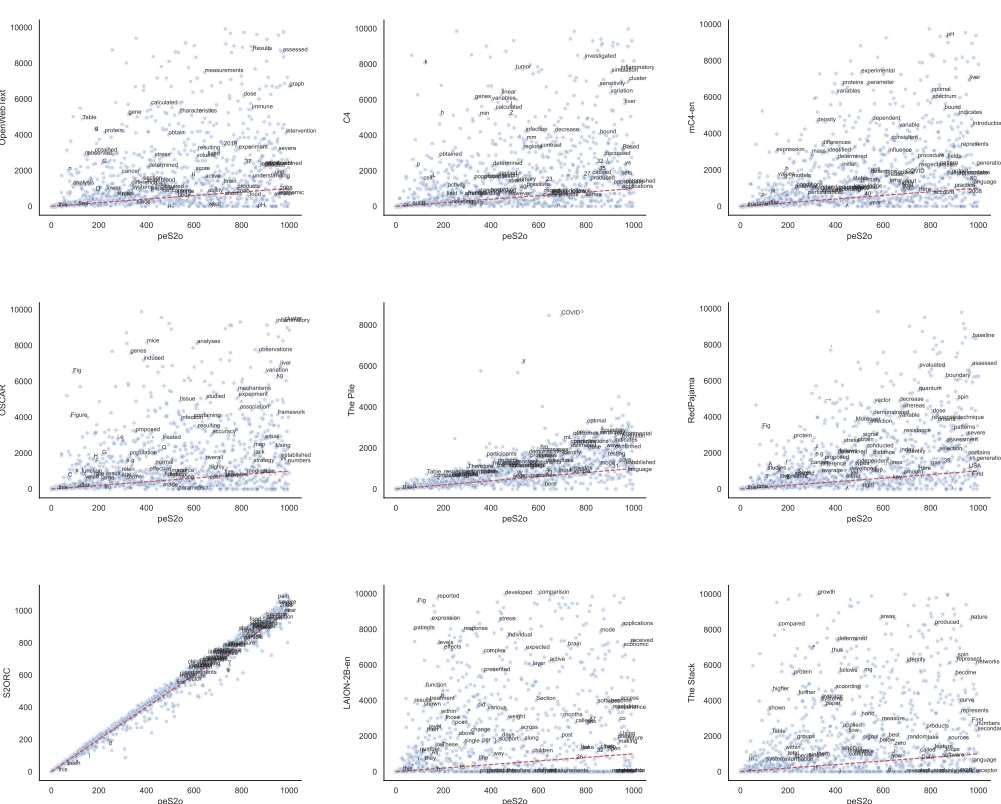

Figure 26: *peS2o* top 1,00 unigrams, and their corresponding indices in the other corpora.

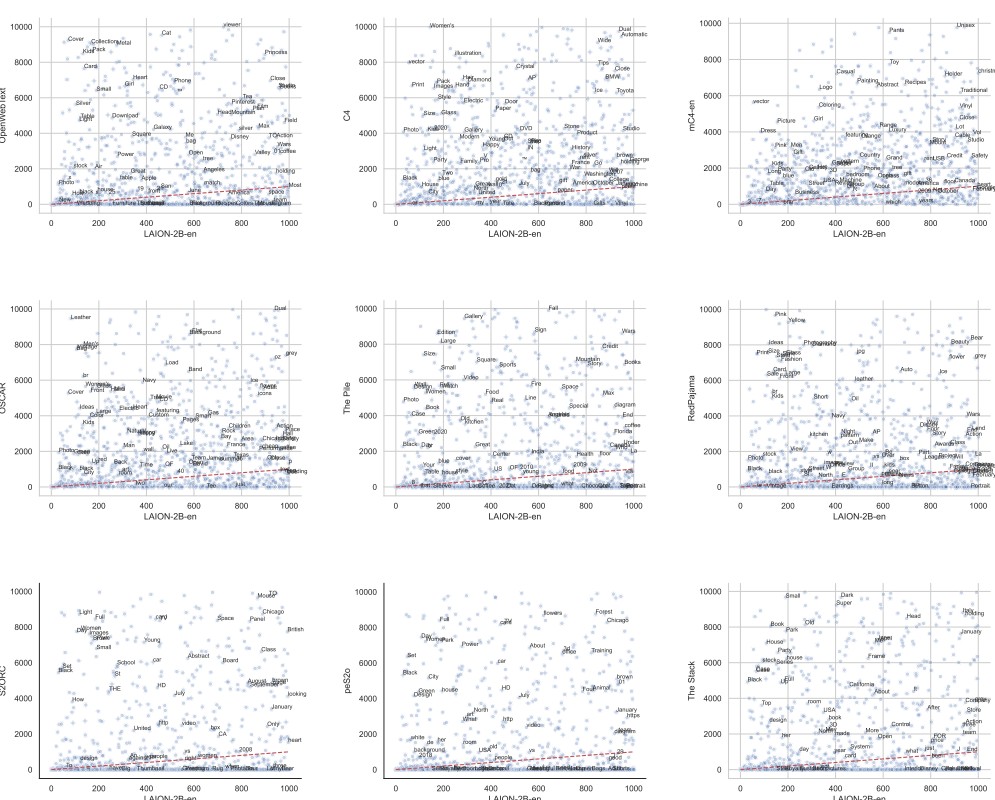

Figure 27: *LAION-2B-en* top 1,00 unigrams, and their corresponding indices in the other corpora.

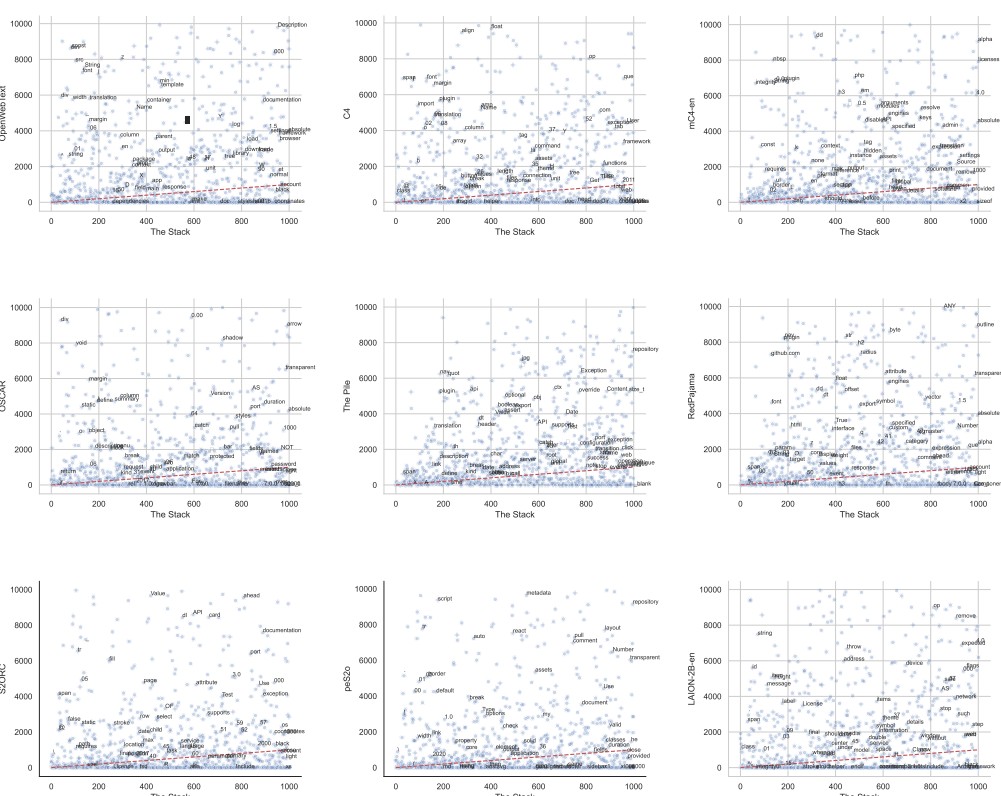

Figure 28: *The Stack* top 1,00 unigrams, and their corresponding indices in the other corpora.

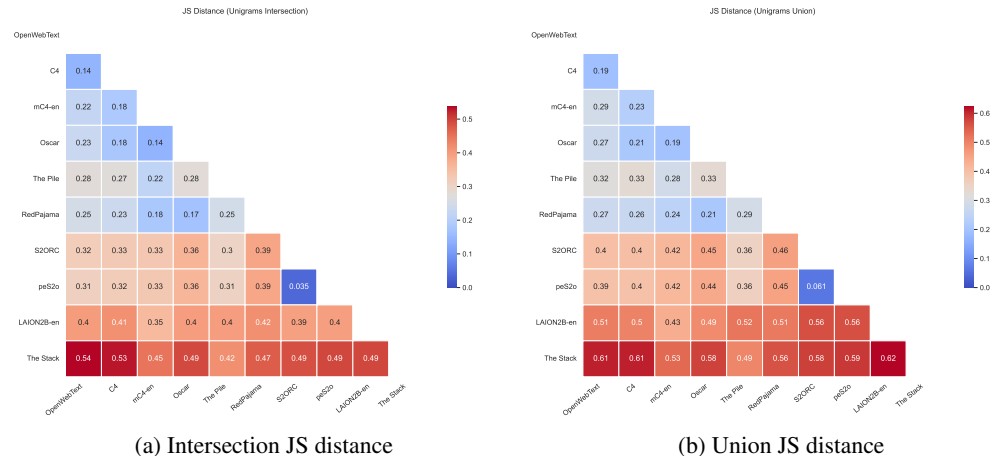

(a) Intersection JS distance        (b) Union JS distance

Figure 29: The Jensen Shannon distance between the top 1,000 most common unigrams in each corpus. The lower the numbers the more similar the corpora are. OpenWebText, C4, mC4-en, OSCAR, The Pile and RedPajama are quite similar to one another (in terms of the common unigrams distribution), and S2ORC, peS2o, LAION-2B-en, and The Stack are quite different from all other corpora.

Table 23: Top 10 exact text overlaps between more than 2 datasets. C4, OSCAR, and RedPajama share the most amount of documents, with over 1.6 million shared documents. Interestingly, even LAION-2B-en, an image-caption corpus overlaps with other corpora, such as C4 and RedPajama (which all share more than 30 thousand documents).

| Corpus Intersection | Count |
|---|---|
| C4 ∩ OSCAR ∩ RedPajama | 1,680,953 |
| C4 ∩ mC4-en ∩ RedPajama | 1,375,088 |
| The Pile ∩ RedPajama ∩ The Stack | 592,364 |
| C4 ∩ The Pile ∩ RedPajama | 118,432 |
| C4 ∩ RedPajama ∩ LAION-2B-en | 30,602 |
| mC4-en ∩ OSCAR ∩ RedPajama | 14,319 |
| C4 ∩ mC4-en ∩ OSCAR | 12,854 |
| C4 ∩ mC4-en ∩ OSCAR ∩ RedPajama | 12,854 |
| OSCAR ∩ The Pile ∩ RedPajama | 6,112 |
| C4 ∩ OSCAR ∩ The Pile | 6,096 |

**Unigram Overlap**     Next, by comparing the 10,000 most common unigrams, we compare the similarity between each corpora pair using the Jensen Shannon distance using (1) the intersection and (2) the union of the two vocabularies. We present the results in Figure 29. On average, we find that *OSCAR*'s unigram distribution is the most similar to all other corpora (0.19 on average). *The Stack*, as expected, is the most distance corpus from all other corpora.

### B.4.2    CORPUS OVERLAP

In this analysis, we compute the overlap between the different corpora, by comparing (1) the texts, and (2) the URLs, when available. The pairwise results are presented in Figure 30 for the texts overlap, and Figure 31 for the URL overlap. We see that text overlap diminishes quickly to zero as more datasets are considered. Table 23 shows the largest text overlaps between more than two datasets. While the largest two are over 1 million document clusters, this is less than 1% of clusters in any of the involved datasets, and overlap size drops rapidly from there. This trend is similar for URL overlaps. The largest 3-corpora overlap is between *C4*, *mC4-en*, and *OSCAR*, with 6,767,877 shared URLS, while the rest of the overlaps share at most a single URL.

We find that documents from *S2ORC* and *peS2o* do not appear in other corpora. While it is likely that some of the academic papers are shared with other corpora, e.g., *The Pile* and *RedPajama*

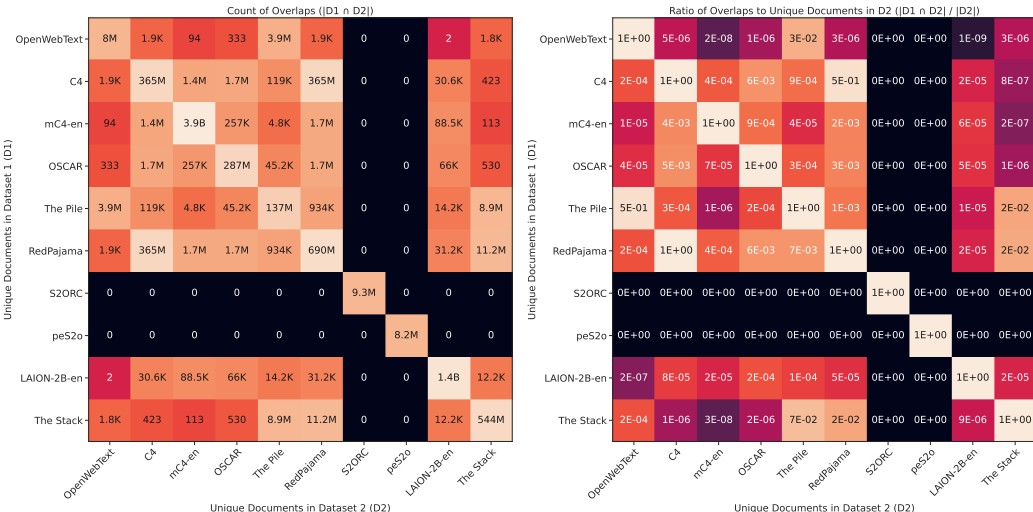

Figure 30: Overlaps of hashed full text between all pairs of datasets as counts and as ratio to dataset size.

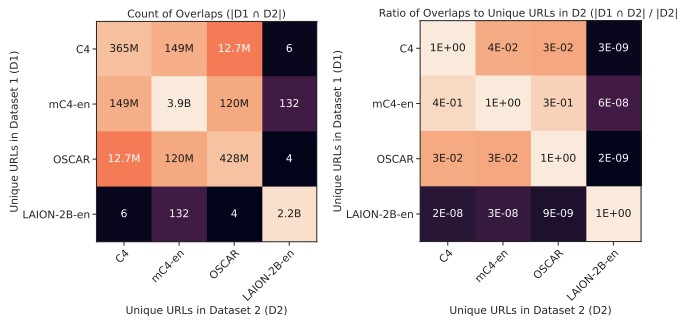

Figure 31: Overlaps of URL string between all pairs of datasets as counts and as ratio to dataset size.

that included arXiv as a data source, there are likely formatting differences that cause the exact string matching to be different. Interestingly, even *S2ORC* and *peS2o* do not contain any exact-text overlapping documents, despite *peS2o* being a cleaned version of *S2ORC*, due to a difference in formatting for parsed paper sections.

While *RedPajama* is 2.5 times larger than *C4* in number of documents and 6.6 larger in number of tokens, we find that 50% of *RedPajama* unique documents originate from *C4*. This can be explained by larger documents (as evident from the largest average document length in *The Stack* of 2,800 tokens per document on average, compared to 420 tokens per document in *C4*, or by duplicate contents of *C4* documents in *RedPajama*. Similarly, 50% of *OpenWebText* unique documents overlap with *The Pile*, which includes *OpenWebText* as a source. Another expected overlap is between datasets with Github as a source (*RedPajama* and *The Pile*), and *The Stack* (which purely consist of Github code).

Finally, we also notice that while *mC4-en* was created from a superset the Common Crawl data used to make *C4*, documents from *C4* only constitute 0.04% of *mC4-en*, while the later is only 10 times larger in size. We speculate that this is due to formatting differences, between the *C4* and *mC4-en* collection.

## C  LIMITATIONS

WIMBD has a few limitations, described below:

- The search tool we use is Elasticsearch. While it is scalable, it was not designed for scaling with large text corpora. In addition, indexing these massive text corpora can take a few days,

and keeping it running is costly. In the future, we hope to explore more cost effective and faster indexing tools.

- Search is currently enabled using Elasticsearch, which only enables exact-match search. Fuzzy, and semantic search are important abilities that we currently do not support.

Table 24: Time benchmark of the different analyses on C4. We ran all of these analyses on a 224-CPUs machine, with 881 Gb memory. * The contamination time was calculated on the test set of COPA, which contains 500 test examples. We also report the estimated cost in dollars based on Google's pricing of the machine we used, that is $9.46 per hour.

| Category | Analysis | Time | Estimated Cost ($) |
|---|---|---|---|
| Data Statistics | Summary Statistics | 6:32 | 1 |
| | Internet Schemas | 2:25 | 0.4 |
| | Internet Domains | 5:38 | 0.9 |
| | Internet Domains per Token | 3:32:07 | 33.4 |
| | Internet Suffixes | 1:56 | 0.3 |
| | Utterance Date Statistics | 2:12 | 0.3 |
| | Geolocation | 1:17 | 0.2 |
| | Language ID | 5:52 | 0.9 |
| Data Quality | Top-1 | 9:08 | 1.4 |
| | Top-2 | 2:14:26 | 21.2 |
| | Top-3 | 5:45:10 | 54.4 |
| | Top-5 | 3:43:58 | 35.3 |
| | Top-10 | 8:43:40 | 82.6 |
| | Top-100 | 3:00:14 | 28.4 |
| | Bot-1 | 18:17 | 2.9 |
| | Duplicates | 8:36 | 1.4 |
| | Length Distribution | 8:56 | 1.4 |
| Comm. Measures | Contamination | *:48 | 0.1 |
| | Toxic Classifier | 3:19:12 | 31.4 |
| | Toxic Taxonomy | 3:15:27 | 30.8 |
| | PII | 24:44 | 3.9 |
| | Demographic Sentiment | 11:41:17 | 110.5 |
| | Total | 46:51:51 | 443.1 |

## D  BENCHMARKING RUNTIMES

This section describes the benchmark times each analysis took to run on the *C4* corpus. While *C4* is not the largest corpora we analyze, it is a popular one, and representative in size. All out analyses were run on a Google cloud compute node with 882GB RAM and 224 CPUs. While the machine is rich in RAM, our analyses typically did not use more than 250GB, and the reason for choosing such machine was the availability of a machine with enough CPU cores, that came along with this amount of memory.

We report the benchmark runs in Table 24. All of the analyses we conducted took less than 12 hours to run, with 13 (out of 22) that took only several minutes, and all of the analyses on *C4* took an estimated of 46 hours and 51 seconds (excluding repeated runs, and the contamination analyses on other evaluation datasets). Note that while the measured time for each run were calculated using the TIME command in linux, there is some variance, and those should be taken as a rough estimate.

We also calculate the estimated costs for each analysis and report it in the same table (Table 24). We use the estimated $9.46 per hour based on https://cloud.google.com/compute/all-pricing for our calculations, making the total cost on *C4* $443.1.[15]

---

[15]This estimation does not include the Elasticsearch hosting costs.

# E    Technical Details

This section describes the algorithms for computing the most common, least common, and total number of unique $n$-grams in a large corpus. Each of these algorithms uses the same trick that was inspired by Bloom filters (Bloom, 1970) as described in section 3.1. As a result these algorithms do not provide exact results, and the accuracy is determined by the amount of memory available for the hash table.

## E.1    Most Common $n$-grams

To collect the (approximate) top-$k$ $n$-grams we start by initializing a hash table of zeros (either u32 or u64) which represent occurrence counts for each $n$-gram, and an empty collection of the top-$k$ $n$-grams. Then we iterate over the $n$-grams in the corpus and for each $n$-gram encountered we take its hash, increment the corresponding count in the hash table, and if that count is at least as large as the current minimum count in the top-$k$ we add that $n$-gram to the top-$k$, potentially evicting another $n$-gram from the top-$k$.

After completing the iteration over the corpus the top-$k$ will be complete and, in the absence of hash collisions, correct. However, the larger the corpus is relative to the hash table, the higher the probability of hash collisions. A large enough corpus will have more unique $n$-grams than there are entries in the hash table, which guarantees hash collisions in the table, leading to inflated counts for some $n$-grams and the potential for false positives in the top-$k$. That's where the accuracy-memory tradeoff comes in. The final counts reported for the top-$k$ $n$-grams will always be an upper bound of the true counts.

## E.2    Least Common $n$-grams

To collect the (approximate) bottom-$k$ $n$-grams we also start by initializing a hash table of u32[16] zeros to represent occurrence counts for each $n$-gram, and an empty collection of the bottom-$k$ $n$-grams. But this time we have to iterate over the corpus' $n$-grams twice.

During the first iteration we tally up the counts just like we do in the top-$k$ algorithm, except that we don't add any $n$-grams to the bottom-$k$ collection. During the second iteration we now already have the final counts of all $n$-grams, so we simply look up the count of each $n$-gram encountered and then add it to the bottom-$k$ collection if its count is low enough, potentially evicting another $n$-gram.

Hash collisions might cause false negatives with the bottom-$k$, i.e. some rare $n$-grams may be missing from bottom-$k$ if they had hash collisions with more frequent $n$-grams. The final counts reported will for the bottom-$k$ $n$-grams always be a lower bound of the true counts.

## E.3    Unique $n$-grams

To estimate the number of unique $n$-grams we initialize a hash table of booleans set to 'false'. Then we iterate over all $n$-grams in the corpus and for each $n$-gram encountered we take its hash and update the corresponding boolean in the table to 'true'. After iterating over the whole corpus we simply have to tally up the number of 'true' entries. This number is the estimate for the number of unique $n$-grams, which will always be a lower bound of the actual number of unique $n$-grams.

---

[16]It's not necessary to use u64 integers when collecting the bottom-$k$ even if there's a possibility of overflow counts, provided overflows are caught and kept at $2^{32}$, since we only care about the exact count of rare $n$-grams which are unlikely to ever reach an overflow.

