# OpenReview forum: "What's In My Big Data?"
_ICLR.cc/2024/Conference — ICLR 2024 spotlight_

### Official Review · Reviewer_jsfy · 2023-10-27

**Soundness:** 4 excellent
**Presentation:** 4 excellent
**Contribution:** 4 excellent
**Rating:** 10
**Confidence:** 5

**Summary:**

The paper introduces an automated system called WIMBD. WIMBD is a framework for processing and analyzing large text corpora, with a variety of evaluation tools. In the work, using WIMBD the quality of ten different corpora widely used train language models (or vision and language models, etc.) has been evaluated. Insightful findings were found and provided, along with a data quality & impact assessment.

**Strengths:**

I think this work is a substantial contribution to the ML community in general, and even broader (like the NLP community). It covers how important good and qualitative data is, above the quantity of the data. As a consequence, I believe it is crucial to adopt tools like WIMDB in order to make sure ML models are trained on good, non-toxic, data in order to avoid reinforcing existing societal biases, and exhibition of toxic language that is negatively impacting as ML models used in day-to-day lives. Systems like WIMDB could also play an important role (as data is) in the interpretability and expandability of ML models' performance + reproducible models.

**Weaknesses:**

Not a weakness per se, but it would have been good to see the authors discuss how scalable the tool is (they tested 10 massively huge datasets) to 100, 1000, etc datasets (with different lengths of magnitudes).

**Questions:**

N/A

---

> ### Author Response · Authors · 2023-11-19
> **Response, clarifications, and discussion**
>
> # Strengths
> Thank you for your thorough review and feedback and for appreciating our *substantial contribution to the ML community*, the *importance* of our work, and our *insightful findings*.
>
> # Clarifications
> **Q1**: “Not a weakness per se, but it would have been good to see the authors discuss how scalable the tool is …”
>
> **A1**: Thank you for the great question. The answer to this question depends on the analysis type.
> Overall, all of our analyses are at most linear with respect to the size of the corpus (for the map-reduce based analyses) and sub-linear after indexing (for the search-based analyses).
>
> We report the run time benchmark and estimated costs of running our analyses on the C4 dataset in Appendix C (Table 23). One can extrapolate the scale based on those.
> Note that for most analyses, the bottleneck is within the number of CPUs, and as such, using more machines or CPUs will scale with more data.
> Interestingly, our duplication analysis, compared to previous work ([Deduplicating Training Data Makes Language Models Better](https://aclanthology.org/2022.acl-long.577.pdf)), requires significantly less RAM and relies mainly on CPUs. This provides a significant advantage since it allows a much better scaling to large-scale datasets with standard machines.
> The main limitation to scale is the search tool, which we used ElasticSearch to index the data. While we can scale to larger datasets, the costs increase proportionally and may become unfeasible. In addition, the indexing times can take several days. In the future, we hope to switch to more scalable search solutions.

---

### Official Review · Reviewer_5Nw1 · 2023-10-31

**Soundness:** 2 fair
**Presentation:** 3 good
**Contribution:** 2 fair
**Rating:** 5
**Confidence:** 4

**Summary:**

The authors present a platform and set of analyses to explore the large-scale datasets used for training LLMs.

**Strengths:**

1. The presentation and writing of the paper is good
2. The insights are presented from multiple large-scale datasets and the proposed methodology could be extended to other large-scale datasets
3. The authors document the set of observations from the dataset that several researchers and practitioners of LLMs in part experience.

**Weaknesses:**

I feel the analyses presented in the paper could be extended or strengthened along multiple lines such as:
1. Can we look beyond the n-gram repetition to measure the duplicity? Is every n-gram duplication bad for training LLMs?
2. What might help to better understand the relevance of the proposed platform is to (re)train the LLMs after removing the unwanted artifacts from the dataset.
3. The impact on the output generated by the LLMs due to such unwanted artifacts in the dataset is still not clear. Some analysis/exploration along this line may help to better motivate the link of work

**Questions:**

See my comments in Weaknesses

---

> ### Author Response · Authors · 2023-11-19
> **Response, clarification, and discussion**
>
> # Strengths
>
> Thank you for your thorough review and feedback and for appreciating the *insights* we reveal on the inspected datasets, the *extendability* of our methodology, the *importance of our analyses*, and the *presentation* of our paper.
>
> We remind the reviewer that our main goal with this work is to create the first platform for understanding the contents of web-scale datasets and to enable a scientific approach to building datasets. This is the first work to provide the same analyses across multiple datasets at this scale. Our work is descriptive, not prescriptive; we provide 16 different analyses across 10 datasets, and changing the datasets across even a subset of the analyses while training models to measure the impact of every change is significantly out of scope.
>
> # Clarifications
>
> **W1a**: “Can we look beyond the n-gram repetition to measure the duplicity? ”
>
> **A1a**: While more sophisticated duplication analysis is interesting and should be explored in future work, even the basic n-gram repetition was not explored across a wide range of datasets and compared consistently before.
>
> **W1b**: “Is every n-gram duplication bad for training LLMs?”
>
> **A1b**: Our work is descriptive, not prescriptive, in providing the first duplication analysis across multiple web-scale corpora. This is the necessary first step to understanding the impact of duplication on trained models. The answer as to whether or not duplication is bad is very complicated and depends on many factors. While it has been explored in a few papers (which we cited in the paper, Section 4.3.2), it is still a topic under debate.
>
> [Deduplicating Training Data Makes Language Models Better](https://aclanthology.org/2022.acl-long.577.pdf)
>
> [Quantifying Memorization Across Neural Language Models](https://openreview.net/forum?id=TatRHT_1cK)
>
> [Pythia: A Suite for Analyzing Large Language Models Across Training and Scaling](https://openreview.net/forum?id=bpRTAnJ8LW)
>
> **W2**: “What might help to better understand the relevance of the proposed platform is to (re)train the LLMs after removing the unwanted artifacts from the dataset.”
>
> **A2**: We agree that studying the effect of the summary statistics we computed can be explored through retraining language models on the modified datasets. However, training LLMs is out of scope for this paper, as it requires computational resources far beyond those used in the WIMBD tooling. Instead, we hope our tooling can help guide research that retrain LLMs.
> Anecdotal evidence that we would like to highlight here is that the most common n-gram analysis was already found useful. Please refer to the general response.
>
> **W3**: “The impact on the output generated by the LLMs due to such unwanted artifacts in the dataset is still not clear. Some analysis/exploration along this line may help to better motivate the link of work”
>
> **A3**: Our work was specifically designed to enable such research and to ask the right questions. We also demonstrated how WIMBD can be utilized through a series of analyses that revealed interesting insights on 10 different datasets.
> We would also like to highlight a few works that have previously explored such questions. While these works had to construct dedicated tools for answering such questions, with WIMBD, these would have been available as an off-the-shelf tool.
>
>
> [Impact of Pretraining Term Frequencies on Few-Shot Numerical Reasoning](https://aclanthology.org/2022.findings-emnlp.59.pdf): Studied the correlation between term frequency of numbers and the ability of models to perform few-shot numerical reasoning.
>
> [The Bias Amplification Paradox in Text-to-Image Generation](https://arxiv.org/pdf/2308.00755.pdf): Studied the effect of bias amplification and compared biases in the training data to biases that models exhibit.
>
> [Speak, Memory: An Archaeology of Books Known to ChatGPT/GPT-4](https://arxiv.org/pdf/2305.00118.pdf): Studied memorization of texts, especially copyrighted texts of models.
>
> [Deduplicating Training Data Mitigates Privacy Risks in Language Models](https://proceedings.mlr.press/v162/kandpal22a.html): Studies how duplicate n-grams affect their frequency appearance in model generation.
>
> Some of these works are also mentioned in the paper at the end of Section 2 (between pages 2 and 3).
> We hope our work will enable many more researchers to perform this kind of work.

---

### Official Review · Reviewer_XCiq · 2023-11-01

**Soundness:** 3 good
**Presentation:** 4 excellent
**Contribution:** 3 good
**Rating:** 8
**Confidence:** 4

**Summary:**

This paper proposes a tool for analyzing the contents of big textual datasets efficiently. It describes the implementation of the platform and gives examples for data statistics and data quality findings from 10 different large corpora for English as a result.

**Strengths:**

- The implemented solution seems efficient and uses scalable methods, making use of simple tools and scaling techniques.
- The proposed tool could be useful to basically any NLP practitioner working with data to be analyzed. It fills a gap of existing tools.
- The shown examples illustrate well how the tool can be used to unearth surprising and previously unknown quality issues and outliers. The appendix has a generous number of analyses and results that might be helpful for anyone working with these datasets.
- The report of benchmark overlap is a nice side finding that will hopefully affect how future models trained on this data are evaluated. It is hard to publish such findings individually, so I am grateful for the authors to include these here and prevent future misinterpretations of results.

**Weaknesses:**

- What the paper doesn’t address is how these instances that are problematic can be efficiently filtered from the data. Would the same tool be able to serve that purpose? Then one could nicely iterate with inspection and cleaning. This is not strictly a weakness as it goes beyond the scope of the proposed tool, but this should at least be discussed to make the observations more actionable.
- The tool is only evaluated on English, and this is not explicitly stated. It is unclear if there are any English-specific biases in the way that e.g. PII regexes are designed, or if the tool can be expected to seamlessly generalize to other languages.
- Given that it is a tool that is supposed to be user-friendly, it would be good to evaluate the usability, as the main contribution of the paper is not a method but a tool. How hard was it to dig out these quality findings, would an average user spot these quickly? Some kind of user study would be good to confirm that the tool is indeed usable. Are the statistics and summaries presented with visualizations to browse? The competitor, Know Your Data, might be good to compare against in that aspect, even if it doesn’t come with the same efficiency.

**Questions:**

- It would be helpful for each inspected corpus to describe briefly what kind of quality guards/filters were employed at its creation (if known), so that e.g. the presence of PII can be interpreted in context. If a corpus is supposed to be PII-filtered already, it would be very surprising to find more PII with this tool - or the regexes used here should be used to make the next generation of PII filter.
- What are the limitations of the platform?

---

> ### Author Response · Authors · 2023-11-19
> **Response, clarification, and discussion**
>
> # Strengths
> Thank you for your thorough review and feedback and for recognizing our *methods' efficient and scalable* nature, their *usefulness to any NLP practitioner* working with data, *filling a missing gap*, showing *surprising and previously unknown issues*, and *providing interesting findings*.
>
> # Clarifications
> ## Weaknesses
> **W1**: "... how these instances that are problematic can be efficiently filtered from the data. … **not strictly a weakness** … but this should at least be discussed …”
>
> **A1**: Thank you for the great comment! We would like to highlight that we briefly discussed this in the Conclusions section in the "Data Curation" paragraph (Section 5). Thanks to your suggestion, we made this point more explicit and extended the paragraph to reflect that.
> WIMBD can serve the purpose of curating data for improving the quality and content of large text corpora.
> Anecdotal evidence is that the most common n-gram analysis was already found useful. Please refer to the general response.
>
> **W2**: "The tool is only evaluated on English, and this is not explicitly stated..."
>
> **A2**: Thank you for highlighting this. We now explicitly state our tool was designed for English in the penultimate paragraph in the Intro (Section 1) and added the following discussion to Section 4.
> Regarding the generalization of the tool and analyses, most of our analyses are not language-reliant. For instance, the most common n-grams or the length distribution. The only analysis that is language-reliant is the toxicity detection, where we use a lexicon or a model trained on English. One would require changing the model or lexicon to reproduce such analysis for other languages.
> Notice, that our tools are easily extendable, and models can be swapped seamlessly.
>
> **W3**: "Given that it is a tool that is supposed to be user-friendly, it would be good to evaluate the usability, …"
>
> **A3**: WIMBD was technically challenging to create due to the scale of the datasets, and we focused on the technical implementation of the analyses, not building a plug-and-play UI for new datasets. In addition, it's designed for use by ML practitioners handling datasets the size of the entire internet. We then demonstrated through extensive experiments that these tools can scale to 10 large-scale datasets.
> Data Measurements Tool and Know Your Data are the main competitors, and, as we wrote in Section 2: "Tools like the Data Measurements Tool and Know Your Data ... focus on smaller datasets since the scale of web data leads to significant technical challenges."
> WIMBD is the first step towards a scientific understanding of how to build web-scale corpora. We hope our analyses are used not just to satisfy curiosity about these datasets's contents but to shape future datasets.
>
> We agree with the comment on the usability of WIMBD for filtering datasets for the creation of future datasets.
> Regarding toxic language, the only dataset we analyzed that explicitly filtered for toxic content (in the form of keyword matching) is C4. Indeed, the matching category from our analysis are the "Offensive-*" categories. Our analysis used a fine-grained lexicon that split this category into "offensive-minority" and "offensive-not-minority". In C4, we only found 47 mentions of the "offensive-not-minority" category, likely due to a difference in the filter used to create C4 and our lexicon. In comparison, other datasets that did not employ such filters contain several million references to such phrases. Interestingly, C4 also has 158K occurrences of the "offensive-minority" category, which were not filtered from the dataset.
> The table showcasing these results is presented in Table 21, and we will add this additional discussion in Appendix B.3.3.
> Regarding PII, none of the datasets we analyzed had an explicit filter for PII.
>
> ## Questions
>
> **Q1**: "It would be helpful for each inspected corpus to describe briefly what kind of quality guards/filters were employed at its creation (if known), so that e.g. the presence of PII can be interpreted in context..."
>
> **A1**: While we agree this would be a great addition, summarizing this information from the *already reported* information from the original papers would be a significant addition. We encourage the interested reader to read the original papers associated with the dataset of interest.
>
>
> **Q2**: "What are the limitations of the platform?"
>
> **A2**: We added a discussion on the limitations to Appendix C. We overview them here as well:
>
> * The search tool we use is elasticsearch. While scalable, it was not designed for scaling with large text corpora. In addition, indexing these massive text corpora can take a few days, and keeping it running is costly. We hope to explore more cost-effective and faster indexing tools in the future.
> * Search is currently enabled using elasticsearch, which only enables exact-match search. Fuzzy and semantic search are important abilities that we currently do not support.

---

> > ### Comment · Reviewer_XCiq · 2023-11-22
> > **Response**
> >
> > Thanks for the clarifications!

---

### Official Review · Reviewer_c8KC · 2023-11-03

**Soundness:** 3 good
**Presentation:** 3 good
**Contribution:** 3 good
**Rating:** 5
**Confidence:** 4

**Summary:**

The premise of the paper is, that we have a poor understanding of the datasets that have been used during the past years to pretrain LLMs. While the curation and preprocessing steps for many popular pretraining datasets have been described, we still lack an understanding on a common set of dimensions with comparable measurements.

This paper proposes a platform that produces a set of dimensions, such as duplication, top-repeated n-grams, domain statistics ect. to characterize a dataset. Throughout the paper, it demonstrates this approach on a set of popular pretraining datasets, such as C4, mC4-EN, Redpajama, OSCAR and others and summarizes findings and provides anecdotes that have been discovered with this approach.

**Strengths:**

The paper is a timely contribution to increase our understanding of data properties that are feeding the LLMs during pretraining. I am not aware of any comparable publication. The techniques used in the approach do not contain any novelty by themselves, but this would not be necessary for the scope of this publication. The paper is easy to follow and well organized.

**Weaknesses:**

While I really like the premise and the execution of the paper, I feel it often falls short of delivering enough insights into the data and could increase its impact. I like the various anecdotes and statistics of those popular datasets, yet many of the dimensions are slightly underdeveloped or lack clarity (see questions).

**Questions:**

- Datasets: Some datasets (e.g. PILE, Redpajama) consist of several datasets, yet it seems the paper only considers the web-crawls in those datasets?
- Token counts: What is a token here? Are you counting whitespace tokens or tokens from a subword tokenizer (which one, the same for all datasets)?
- Domain counts: As the text tells us "C4 contains documents from a diverse set of domains, and even the percentage of the most common one, patents.google.com, is less than 0.05%. [...]. Similarly, arxiv.org is responsible for more than 12% of the documents in RedPajama." While this is interesting, it feels like this topic is just scratching the surface and its still unclear how the rest of the data looks like.
- N-Grams: Over tokens or characters?
- Document duplication: The text says MD5, but then refers to "Compressed counts", which seems to describe MinHash?
- Benchmark contamination: Frankly, the Llama2 technical report set a high bar for investigating contamination of relevant benchmarks. The selection here is not helpful for SOTA research, and the selection criteria is questionable: Why are benchmarks with less than one input field discarded? Why Promptsource? This part should either left out or fleshed out considerably with more details.

---

> ### Author Response · Authors · 2023-11-19
> **Response, clarification, and a new experiment**
>
> # Strengths
> Thank you for your thorough review and feedback and for recognizing our work is *timely*, *fills in an essential gap* in the field for *increasing our understanding of dataset content*, and that it is *well-written and easy to follow*.
>
>
> # Clarifications
>
> ## Weaknesses
>
> **W1**: “falls short of delivering insights”
>
> **A1**: We agree that there are many more interesting analyses to perform on data. However, we are, in fact, the first work that conducted a large-scale study on datasets at this scale and across different datasets. Our efforts will open the door to many more interesting questions and conduct better scientific research in this field.
> In addition, these analyses are extendable throughout our to-be-public framework, enabling other researchers and practitioners to run their own analyses.
> Please also refer to the appreciation of the other reviewers (XCiq, 5Nw1, jsfy) for the insights our work has enabled.
>
> **W2**: "lack clarity"
>
> **A2**: Please see the answers below. We would love to clarify additional details and provide further insights that you might find interesting.
>
> ## Questions
> **Q1**: “Datasets: ... it seems the paper only considers the web-crawls in those datasets?”
>
> **A1**: Each dataset in our analysis (including The Pile and RedPajama) is analyzed in full; we do not only analyze the web-crawled data.
> The only corpora where we explore a subset are the ones that involve non-English text, specifically - LAION and mC4, where we look at the English subset. We clarify this in the paper at the beginning of Section 4.
>
> **Q2**: “Token counts: What is a token here?”
>
> **A2**: Throughout the paper, we use the Unicode text segmentation tokenizer, which we state in footnote 1 on page 4.
> Also, note that WIMBD allows the use of any tokenizer that is supported by the HuggingFace tokenizer library.
>
> **Q3**: “Domain counts: … While this is interesting, it feels like this topic is just scratching the surface and its still unclear how the rest of the data looks like.”
>
> **A3**: Thank you for this suggestion. We agree that a thorough analysis of the domains is an exciting opportunity!
> We also agree that the source domains contain rich metadata. Our paper has many analyses in the appendix, including the distribution over internet domains (B.1.1.), internet domain schemes (B.1.2.), internet domain suffixes (B.1.3.), and estimated geolocation (B.1.5.).
>
> In addition, we added a new analysis to the paper (Table 7 and the discussion to Appendix B.1.1). Here, we compute the quantiles of the number of tokens per domain and then report the 1, 25, 50, 75, and 99 quantiles and the number of unique domains in each corpus. We observe that 50% of the documents in C4 have 964 or fewer tokens, and 25% of the documents in LAION have 6 or fewer tokens.
>
> Note that we will also release all of the artifacts, including the domain distribution. As such, researchers can perform additional analyses on this data.
>
>
> **Q4**: “N-Grams: Over tokens or characters?”
>
> **A4**: All n-grams are computed over tokens. We mentioned this in footnote 1 on page 4 but added a clarification at the beginning of Section 4.3.1.
>
>
> **Q5**: "Document duplication: The text says MD5, but then refers to "Compressed counts", which seems to describe MinHash?"
> **A5**: We have different ways of hashing the original objects (here, entire documents), all of which we refer to as "compressed counts". In this case, we use the MD5 hash function.
>
> **Q6**: "Benchmark contamination: … This part should … [have] more details."
>
> **A6**: Due to the lack of space, we omitted some details, but we added it to the revised paper, and we describe our thinking process here as well.
> We acknowledge that this is an important question.
> We would also like to highlight that this is the first work conducting such extensive evaluation across different datasets for comparable measurements.
> We agree that additional experiments are interesting and should be answered in the future, which we leave to future work (that can be performed with WIMBD).
>
> **Elaboration**
>
> We looked for a large-scale benchmark that can be quickly processed and parsed. In addition, due to the scale of this experiment, we made several assumptions about what can be counted as contamination. Due to the nature of paired input tasks, the occurrence of both inputs in the same document increases the likelihood of these texts being part of a contaminated example. In contrast, datasets that involve a single input may originate from naturally occurring text from the internet, which was then labeled to create a dataset. As there is no taxonomy of such datasets, this involves manual annotation of hundreds of datasets, which we were trying to avoid in this version.
> We discussed some of these decisions in the original version, in Appendix B.3.1, and added this additional discussion under the new paragraph: "Rationales of the Design Choices".

---

### Author Response · Authors · 2023-11-19
**General Author Response**

### Thank You

We want to thank all reviewers for the detailed reviews and time put into reviewing our paper.

### Strengths Summary

We would also like to highlight that the reviewers understand how *little understanding there is on pretraining datasets* (c8KC) and how our paper proposes a platform to *standardize pretraining corpora summarization* (c8KC) that is *extendable and efficient* (XCiq). In addition, it is *timely* (c8KC), there are *no other comparable works* in the field (c8KC, XCiq), and it has *substantial contributions to the ML community and beyond* (c8KC, jsfy). Our analyses provide *surprising and novel findings* (XCiq, jsfy) on popular and web-scale text corpora.


### WIMBD's impact
One of the repeated questions from reviewers was about the implications of WIMBD and its impact. We want to share anecdotal evidence of WIMBD's impact that was privately shared with us.
The most common n-gram analyses revealed interesting insights into a new corpus under development. The creators' confidentiality informed us it was beneficial for them in the curation process of their dataset, as they found a lot of low-quality data in the original data and could detect and filter it using WIMBD. While we cannot reveal additional information at the moment to maintain anonymity, we can make this information available in the camera-ready.


### Detailed Replies
We provide detailed replies to each reviewer individually, as well as clarifications and new experiments. We updated the paper with additional clarifications and experiments.

---

### Meta-Review · Program_Chairs · 2023-12-03

**Metareview:**

This paper introduces a set of methods/toolkit to analyze corpora used to pre-train language models, and illustrates how the toolkit can be used to uncover various properties of widely used corpora, such as the amount of duplicate content.

While the used methods might not be technically novel, their assemblage in a coherent practical framework is immensely useful.

The weaknesses pointed out by two reviewers are partially addressed in the rebuttal, and they seem in any case more like caveats that should be added to the paper or possible directions for further work, than real flaws.

**Justification For Why Not Higher Score:**

Some concerns remain.

**Justification For Why Not Lower Score:**

The paper proposes a practical and effective approach to assess the quality of the corpora used to pre-train LLMs. This is an extremely useful contribution, and it should be given the best platform the conference can offer.

---

### Decision · Program_Chairs · 2024-01-16

Accept (spotlight)